# Prognostic and Health Management of Critical Aircraft Systems and Components: An Overview

**DOI:** 10.3390/s23198124

**Published:** 2023-09-27

**Authors:** Shuai Fu, Nicolas P. Avdelidis

**Affiliations:** IVHM Centre, School of Aerospace, Transport and Manufacturing, Cranfield University, Bedford MK43 0AL, UK; np.avdel@cranfield.ac.uk

**Keywords:** prognostics and health management, hybrid model, remaining useful life, physics-based model, data driven model, aircraft systems, condition-based maintenance, predictive

## Abstract

Prognostic and health management (PHM) plays a vital role in ensuring the safety and reliability of aircraft systems. The process entails the proactive surveillance and evaluation of the state and functional effectiveness of crucial subsystems. The principal aim of PHM is to predict the remaining useful life (RUL) of subsystems and proactively mitigate future breakdowns in order to minimize consequences. The achievement of this objective is helped by employing predictive modeling techniques and doing real-time data analysis. The incorporation of prognostic methodologies is of utmost importance in the execution of condition-based maintenance (CBM), a strategic approach that emphasizes the prioritization of repairing components that have experienced quantifiable damage. Multiple methodologies are employed to support the advancement of prognostics for aviation systems, encompassing physics-based modeling, data-driven techniques, and hybrid prognosis. These methodologies enable the prediction and mitigation of failures by identifying relevant health indicators. Despite the promising outcomes in the aviation sector pertaining to the implementation of PHM, there exists a deficiency in the research concerning the efficient integration of hybrid PHM applications. The primary aim of this paper is to provide a thorough analysis of the current state of research advancements in prognostics for aircraft systems, with a specific focus on prominent algorithms and their practical applications and challenges. The paper concludes by providing a detailed analysis of prospective directions for future research within the field.

## 1. Introduction

The maintenance of the safety and dependability of aircraft heavily relies on the prognostic and health management (PHM) of essential subsystems or components [1]. Predictive maintenance (PM) solutions rely on the utilization of real-time data to diagnose potential failures and forecast the overall health of machinery. The process is distinguished by its proactive aspect, necessitating the application of predictive modelling tools to trigger maintenance operations, and its capacity to anticipate probable faults before they actually happen [2,3]. PHM is a systematic approach employed to monitor and assess the condition and operational efficiency of essential subsystems or components inside an aviation system. The primary objective of PHM is to uphold the integrity and reliability of the aircraft by proactively forecasting and averting malfunctions prior to their manifestation. PHM systems play a crucial role in aviation maintenance by offering diagnostic and prognostic capabilities. These systems take advantage of the abundant sensor data available on contemporary aircraft [4,5]. Several examples of PHM methodologies encompass data analysis, modeling, and simulation techniques. The utilization of these tactics enables the anticipation and mitigation of failures in advance of their actual occurrence.

The estimation of the remaining useful life (RUL) of subsystems is a crucial aspect of PHM in aviation. This technique is of considerable significance for several reasons. First and foremost, this enables operations to enhance their maintenance strategies by transitioning from predetermined timetables to proactive, condition-based methodologies. As a result, this subsequently leads to a decrease in unplanned periods of inactivity and a reduction in expenses associated with maintenance. Furthermore, the forecast of RUL plays a significant role in improving safety and dependability, particularly in industries with high safety requirements like aviation. The proactive identification of possible failures or deterioration in subsystems significantly contributes to the prevention of accidents and operational interruptions. Moreover, it facilitates the process of making decisions based on data, thus offering significant insights into the health and performance of subsystems. Consequently, this facilitates the ability to make more knowledgeable decisions pertaining to the upkeep, fixing, and substitution.

Due to its ability to forecast the RUL of a system while in operation, PHM facilitates the implementation of condition-based maintenance (CBM), a novel maintenance approach that exclusively addresses the repair or replacement of components that have incurred real damage. This method has the potential to diminish the overall life cycle costs associated with maintenance. CBM encompasses a set of hardware and software systems that are automated in nature. These systems are designed to effectively monitor, identify, isolate, and anticipate the performance and deterioration of equipment. Importantly, CBM achieves these objectives without causing any interruptions to the everyday operation of the systems in question. CBM is a maintenance approach that relies on the current state of equipment or components, as opposed to relying on breakdown or planned repair. Prognostics plays a crucial role as an enabling technology for CBM, facilitating the timely decision-making process for maintenance activities by offering a range of the following advantageous outcomes:Early Fault Detection: PHM systems possess the capability to scrutinize data obtained from diverse sensors and discern minute alterations in the health of assets. This enables the timely recognition of prospective difficulties prior to their escalation into major problems;The concept of PHM involves the ability to anticipate the failure or maintenance needs of an asset, allowing for the proactive scheduling of maintenance operations. This approach coincides with the ideas of CBM;Data-driven decision-making is a process in which PHM systems utilize data analytics and machine learning techniques to provide valuable insights into the health and performance of assets. These insights enable maintenance teams to make well-informed decisions;The use of PHM enables the enhancement of maintenance plans through the prioritization of assets that exhibit a higher likelihood of failure or those that would obtain the greatest value from maintenance activities;The use of PHM may effectively mitigate unexpected downtime and production interruptions by promptly identifying concerns and strategically planning maintenance activities.

Figure 1 illustrates the operationalization of prognosis within a conceptual framework for Open System Architecture Condition Based Maintenance system (OSA-CBM).

Every system has a decline in performance when it operates under stress or load over a period of time. Hence, it is important to implement maintenance practices to ensure a desirable degree of dependability over the lifespan of the system. The initial kind of maintenance that was practiced is known as corrective maintenance, often referred to as reactive, unplanned, or breakdown maintenance. This type of maintenance is performed solely in response to a failure that has already happened, hence it may be characterized as passive in its approach. The practice of corrective maintenance, sometimes referred to as first-generation maintenance, has been implemented since the inception of machines created by humans. Given that corrective maintenance takes place once the system has fully exhausted its useful life, there is no prior preparatory period for maintenance activities. The duration of maintenance is significantly prolonged, and the associated costs of forced outages are maximized, unless there is an existing availability of replacement components. Due to the inherent difficulty in accurately forecasting system failures, the system’s availability is notably suboptimal. Nevertheless, the replacement process exclusively targets components that have undergone deterioration, resulting in a minimal quantity of replacement parts [6].

**Figure 1 sensors-23-08124-f001:**
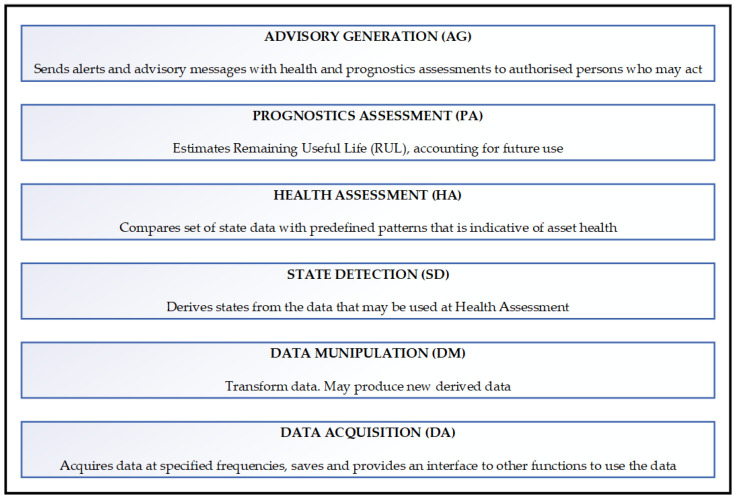
The OSA-CBM (ISO 13374 [7]) functional block diagram (Source Mimosa).

One subsequent maintenance strategy is time-based preventive maintenance, also known as scheduled maintenance or second-generation maintenance. This approach establishes regular intervals for maintenance activities to prevent failures, irrespective of the current health condition of the system. Traditional approaches for predicting dependability often rely on either handbooks or previous field data. The maintenance method that is widely adopted and involves the scheduling of most replacements in advance is the most popular. The primary consideration in preventative maintenance is cost, as it entails the replacement of all components, even though a significant portion of them may not require replacement. The cost-effectiveness of preventive maintenance is contingent upon the assumption that all components are anticipated to break at around the same time. Nevertheless, this approach proves to be efficient only in cases when there is a limited occurrence of part failures, as it necessitates the replacement of several parts that are not anticipated to fail.

To elucidate the matter of inefficient maintenance practices, the maintenance procedures involved in rectifying fractures present in airplane panels is explained as follows: according to the regulations set out by the Federal Aviation Administration (FAA), it is mandatory to address any cracks measuring 0.1 inches in size during a type-C inspection, which is conducted at intervals of 6000 flight cycles [8]. The purpose of this rule is to ensure the safety of the airplane frame by conducting a reliability evaluation. The objective is to achieve a reliability level, indicating that there should be no more than one failure every ten million instances. If a fracture with a size of 0.1 inches is present, the likelihood of the crack expanding and becoming unstable within the following 6000 flights is estimated to be around 10^−7^. Hence, in the event of a type-C inspection, the identification of a fracture measuring 0.1 inch necessitates repair, whereas the detection of many cracks necessitates panel replacement.

The maintenance cost increases as contemporary systems grow increasingly sophisticated and maintain greater levels of dependability due to the fast development of technology. Over time, the implementation of PM has emerged as a significant financial burden for several industrial enterprises. The implementation of CBM has emerged as a viable approach to save maintenance expenses while ensuring the desired standards of dependability and safety. CBM involves performing maintenance activities solely when necessary, and PHM serves as the pivotal technology to achieve this objective. CBM exhibits notable distinctions from conventional maintenance methodologies such as PM and reactive maintenance (RM). In contrast to conventional systems that rely on predetermined timetables or reactive responses to failures, CBM emphasizes proactive maintenance based on real-time condition assessments. This paper aims to elucidate the distinctions between CBM and conventional maintenance techniques while also exploring the role of PHM in supporting CBM. In essence, CBM distinguishes itself from conventional maintenance methodologies by its emphasis on continuous asset condition monitoring and subsequent maintenance actions prompted by the observed condition. The integration of PHM with CBM is advantageous due to its capacity to use data-driven insights, detect faults at an early stage, and enable PM. As a result, this integration contributes to the enhancement of asset dependability, cost reduction, and extension of asset lifespan. Table 1 presents a comparative comparison of CBM and conventional maintenance approaches.

In the process of developing commercial aircraft, many models are employed to facilitate the design of PHM systems, or alternatively, as integral components of the PHM itself. These models may encompass physics-based modeling, sensitivity analysis, and uncertainty propagation [9,10]. The objective of these models is to anticipate and avert failures prior to their occurrence through the identification of the most pertinent health indicators (HIs) and the estimation of probability density functions (PDFs) for HIs in both optimal and deteriorated conditions [9,10].

Within the realm of evaluating aviation system performance, the utilization of physics-based modeling entails the application of comprehensive understanding of the system to construct models capable of forecasting and preempting faults before to their occurrence. These models have the potential to be utilized in conjunction with other methodologies, such as data-driven modeling and physics-based models, to enhance their precision and dependability [11].

To offer a comprehensive assessment of the current research landscape on PHM within the aviation industry, authors of this paper conducted a systematic search utilizing pertinent keywords such as ‘prognostic’, ‘aircraft’, and ‘system’ in the SCOPUS research database. The results suggest that the implementation of PHM in aircraft systems has produced favorable results, as demonstrated by the considerable volume of published research work. Nevertheless, it is crucial to acknowledge that a notable discrepancy persists in the research advancements of PHM for aircraft systems compared to other associated fields. The extent of the research gap is more evident when examining the keywords ‘hybrid prognostic’, ‘aircraft’, and ‘system’. This suggests that there remains a substantial amount of work to be performed in effectively integrating hybrid PHM applications in the aviation sector. The searched results from 1972 to 2024 are depicted in Figure 2 and Figure 3, which are investigated initially by SCOPUS. 

This paper aims to provide an overview of the current research status pertaining to the application of PHM in the aviation industry. It accomplishes this by showcasing various mainstream algorithms and their applications. The intended audience for this paper includes researchers, academicians, and engineers seeking a comprehensive understanding of PHM in the aviation industry. 

The subsequent sections of the paper are structured in the following manner. Section 1 provides an overview of prediction approaches and their connection to integrated vehicle health management (IVHM). It also introduces the concept of prognostics and concludes by discussing the issues associated with prognostics. Section 2 provides an overview of prognostics approaches in physics-based models and data-driven models. It discusses the fundamental concepts, applications, and issues associated with these approaches in the context of aircraft systems. Section 3 of this paper presents a contemporary and promising amalgamation of hybrid prognostic methodologies that have been combined with both physics-based and data-driven models. Section 4 discusses the current challenges in the field. Section 5 serves as the concluding section of the paper, providing an overview of the main findings and implications of the study. Additionally, it highlights potential directions for future research and further exploration in the field.

### 1.1. Types of Prediction Techniques

Predictions can be realized using several methodologies, such as statistical analysis, experience-based methods, computer models, physic-of-failure (PoF) approaches, or combinations of these techniques [12].

The field of statistics is related to the collection, analysis, interpretation, presentation, and organization of data. This methodology employs many statistical techniques, including autoregressive moving average (ARMA) and exponential smoothing, to analyze data. These techniques utilize random variables to enhance the distribution of unknown characteristics in newly acquired data. Regression analysis is a statistical method used to establish the relationships between variables and estimate the parameter values to make predictions about the RUL. ARMA is commonly employed in typical operational scenarios to discern and comprehend the dynamic characteristics of various components. When utilized in the context of lifecycle issues, this model has demonstrated its ability to generate precise and dependable estimates of RUL;The use of this strategy is predicated upon the discernment of individuals with specialized expertise. Knowledge may be categorized into the following two forms: explicit and tacit. It is acquired via the expertise of those who possess a deep understanding of a particular field. This approach is employed to facilitate decision-making related to the maintenance of deterioration, whereby ongoing monitoring of processes and objects is conducted. Understanding is derived from the collection of data acquired from both failed occurrences and developmental test events. The examination of the data allows for the identification of characteristics derived from degradation mechanisms, which in turn aids in the creation of datasets. Additionally, it enables the implementation of classification criteria to ascertain the RUL of an asset by establishing a predetermined threshold level;Computational intelligence, sometimes referred to as computer model, encompasses the utilization of fuzzy logic and neural networks that rely on parameters and input data to generate the intended output. Artificial neural networks (ANNs) utilize data obtained from continuous monitoring systems and necessitate the presence of training samples. ANNs, sometimes referred to as ‘black boxes’, offer limited visibility into its internal mechanisms [13]; however, by using ANNs, data obtained from sensors may be processed to forecast the RUL of an asset. Alternative methodologies include Bayesian prediction and support vector machines, both of which employ statistical techniques to estimate conditions based on limited samples to establish a foundation for predictive learning;The PoF methodology necessitates the use of parametric data and encompasses several methodologies, such as continuum damage mechanics, linear damage rules, nonlinear damage curves, and two-stage linearization. Methods for modifying the life curve of stress and load interaction, as well as concepts related to fracture development and energy-based damage models, are also accessible;Fusion refers to the process of combining several sets of data to create a more refined and consolidated state. The proposed methodology involves the extraction, pre-processing, and fusion of data to achieve an accurate and efficient forecasting of the RUL. One potential approach to enhance the integration of fusion is to employ the fuzzy method to categorize data, hence augmenting the precision of the RUL estimation. In the realm of uncertainty surrounding RUL estimation, the integration of on-demand data obtained from various sensors is accomplished using centralized or decentralized methods to achieve precise predictions of useful life. This fusion process is facilitated by the utilization of principal component analysis.

### 1.2. Prognostics and Integrated Vehicle Health Management (IVHM)

The inception of PHM can be traced back to the 1980s when the Civil Aviation Authority (CAA) of the United Kingdom initiated its implementation with the aim of mitigating the occurrence of helicopter accidents. Subsequently, in the 1990s, PHM underwent further advancements by including health and usage monitoring systems (HUMS), which enable the measurement of both the health conditions and performance of helicopters. The implementation of the HUMS has yielded significant outcomes in the reduction of accident rates, surpassing a 50% decrease. This fatal accident rate is illustrated in Figure 3, which demonstrates the effectiveness of HUMS when applied to in-service helicopters.

In the 1990s, the aerospace research division of NASA in the United States included the idea of vehicle health monitoring (VHM), which involves the monitoring of the health status of outer space vehicles. Nevertheless, it was subsequently substituted with a more comprehensive designation known as IVHM or system health management (SHM), which encompasses the prognostics of diverse space systems [14]. During the early 2000s, the Defense Advanced Research Projects Agency (DARPA) in the United States of America successfully devised two systems, namely, the Structural Integrity Prognosis System (SIPS) and Condition-Based Maintenance Plus (CBM+), both serving a similar objective. The term prognostics and health management were initially introduced in the program for joint strike fighter (JSF) development (Joint Strike Fighter Program Office, 2016) [15].

Subsequently, the technology known as PHM has experienced substantial advancements across several domains. These advancements encompass the comprehensive examination of failure physics, the refinement of sensor technologies, the extraction of relevant features, the implementation of diagnostic techniques for the detection and classification of faults, as well as the establishment of prognostic methodologies for the prediction of failures. These strategies have been extensively investigated and developed upon in several sectors. The proliferation of technological advancements in the business has led to a growing body of literature that explores successful applications across several sectors [16]. 

The development of IVHM arose from the recognition that PM has mostly concentrated on individual aircraft subsystems in isolation from one another. The manufacturers of the engines, avionics, structure, and other components designed their own PHM systems independently. IVHM proposes that PHM should be implemented as a comprehensive and unified platform, backed by a solid commercial rationale. The design and construction of an IVHM system should adhere to an open and layered architecture, employing a systems-engineering approach to achieve comprehensive platform capabilities. This system should serve as a foundation for improving or substituting conventional maintenance practices, hence providing maintenance credits. The utilization of an open architecture allows manufacturers of subsystems to construct PM systems that possess the capability to exchange data with other platform systems. While the concepts of IVHM have mostly been formulated within the aerospace sector, the underlying principles may be applied to many other industries as well. In various scenarios, there may arise a necessity to broaden the interpretation of the term vehicle within the context of IVHM, so as to encompass any form of industrial facility. The term vehicle is sometimes misinterpreted as exclusively referring to movable assets, which is an inaccurate assumption. For a more comprehensive understanding of IVHM, one may refer to [13] (Figure 4). 

Numerous industrial facilities already employ predictive maintenance systems, which encompass various techniques such as periodic vibration analysis via portable vibration sensing equipment, oil and oil debris analysis conducted through laboratory testing, and perhaps additional non-destructive testing (NDT) or examination (NDE) methods. Typically, these findings are assessed independently from one another and without considering any other PM system. Frequently, the information technology (IT) systems employed for data collection and result generation operate independently, with the data being private and hence not amenable to sharing across disparate IT systems. This overlooks the potential to integrate the available information to obtain an understanding of machinery health and condition, in accordance with the principles underlying IVHM. 

### 1.3. Challenges in Prognostics

While the field of PHM offers several advantages, presents numerous benefits, it is crucial to recognize that there are still obstacles that need to be addressed [17]. 

The implementation of optimal sensor selection and localization is a key. The process of acquiring data is an initial and crucial part of prognostics. The measurement of environmental, rational, and performance parameters of a system frequently necessitates the utilization of sensor systems. The prognostic performance may be compromised due to inaccurate readings resulting from the inappropriate selection and placement of sensors. The sensors must possess the capability to precisely quantify the alterations in the parameters associated with the catastrophic failure mechanism. It is important to consider the potential for sensor reliability and failures. Several solutions have been proposed to enhance the reliability of sensors. These include the utilization of redundant sensors to monitor a given system and the implementation of sensor validation techniques to evaluate the accuracy and reliability of a sensor system, afterwards making appropriate adjustments or corrections.

Feature extraction is a crucial stage in prognostics because it enables the collection of data that is directly related to the occurrence of damage, thereby ensuring the significance of the prognostic analysis. However, in a number of instances, the collection of damage data is instantaneously difficult or impossible. Due to the continuous rotation of the bearing, measuring the fractures in the bearing’s race presents a significant challenge. In this scenario, damage estimation is accomplished by measuring a system reaction that is directly related to the damage. The installation of accelerometers near bearings to monitor and evaluate the magnitude of vibration signals is an example of an accelerometer’s application. Due to the presence of noise in system vibration, it becomes difficult to extract damage-related signals. In the context of complex systems, the amount of harm is not negligible in comparison to the entire system. The magnitude of the signals associated with damage is typically much less than the magnitude of the signals associated with the system’s response. Consequently, it is difficult to extract these small signals associated with damage from a relatively large quantity of noise.

In line with what has been mentioned so far, most people agree that there are the following two main types of prediction methods: those based on physics and those based on data. To figure out when a system has reached the end of its useful life, physics-based methods use an understanding of failure mechanism models or other models that describe the system. Few data points are required to consistently predict the RUL, which is a significant advantage; however, it is essential to understand the failure process. In the context of fracture development models, it is essential to consider a few factors, including the properties of the materials used, the geometry of the structure, how it is utilized, and how it is loaded by the environment. Incorporating these features into systems may be difficult. In addition, the use of models requires a comprehensive understanding of the fundamental physical mechanisms for failure and their operational principles. In complex systems, however, it is difficult to obtain such models, so physics-based methods must account for significant limitations. Data-driven strategies employ empirical data from the actual world to acquire knowledge and ascertain the underlying factors contributing to deterioration. This enables the prediction of future states without relying on explicit physical models. Therefore, the precision of RUL prediction outcomes mostly relies on the caliber of the training dataset. This paper provides a comprehensive examination of the characteristics of prevalent algorithms employed in physics-based and data-driven methodologies. The objective is to facilitate the development of hybrid methodologies by comprehensively examining the characteristics of each algorithm.

The analysis of prognostic uncertainty and the assessment of its accuracy pose significant challenges. One significant obstacle in the implementation of prognostics is to the formulation of approaches that may effectively address the uncertainty encountered in practical situations, resulting in less precise prognostications. Figure 5 illustrates many sources of uncertainty that are found in the field of prognostics. These sources are often classified into the following three distinct categories: 

The presence of these uncertainties possesses the capacity to give rise to significant disparities between projected results and the factual condition. The significance of developing approaches that can accurately describe uncertainty bounds and confidence levels for prognosis cannot be overstated. In order to ascertain and measure the level of trust in a prognostics system, it is important to incorporate a methodology for assessing the accuracy of prognostications. There is currently no agreement among scholars on the most suitable and universally recognized methods for assessing prognostic performance; however, some academics have put out several techniques for consideration. The authors [18] offered a comprehensive array of metrics aimed at assessing the efficacy of prognostics algorithms in their study. The criteria under consideration encompass prognostics hit score, false alarm rate, missed estimation rate, accurate rejection rate, and prognostics effectivity. In addition, reference [19] developed a comprehensive set of measurements for examining the essential components of RUL prediction. The metrics include prognostic horizon, α–λ performance, relative accuracy, and convergence, as seen in Figure 6.

The aviation industry has several challenges when it comes to the application of PHM. One major impediment is the integration of PHM systems into current aircraft and maintenance operations. The issue under consideration covers the following several unique aspects:Compatibility with legacy systems: a considerable proportion of aircraft now in service were manufactured and designed prior to the extensive use of PHM technology. The integration of PHM systems into outdated aircraft presents difficulties as a result of potential inconsistencies with the preexisting onboard systems and data connection protocols, which may not have been initially developed to support PHM capabilities. The implementation of PHM capabilities in these aircraft might potentially result in substantial costs and need complex processes;The aggregation of data acquired from various sensors and systems deployed on the aircraft is a crucial component of PHM, highlighting the significance of data integration in this field. The process of accurately gathering, transferring, and consolidating data from several sources into a unified PHM system is a significant challenge. The successful attainment of this aim requires the use of standardized data formats and communication protocols;The aviation sector is widely recognized as critical infrastructure owing to its significant role in upholding essential social functions. In this particular industry, the data generated by PHM systems possesses considerable sensitivity and necessitates the implementation of strong cybersecurity protocols. The preservation of the confidentiality and integrity of sensitive data in the face of cyber threats is of paramount significance. The integration of PHM into aviation systems poses a significant challenge in terms of maintaining cybersecurity. This challenge requires the establishment of robust cybersecurity standards and continuous monitoring;The aviation industry is subjected to comprehensive laws, which require strict compliance with rigorous safety and reliability standards when introducing new technologies and systems. The endeavor of assuring the adherence of PHM systems to aviation standards may provide obstacles in terms of intricacy and time expenditure;The discipline of human factors recognizes that PHM systems yield a significant amount of data and diagnostic information. It is crucial to guarantee the proper dissemination of the aforementioned information to pilots, maintenance crews, and other pertinent stakeholders in a manner that is easily understandable and can be promptly implemented. The successful implementation of PHM requires the meticulous consideration of human factors, encompassing elements such as user interface design and training;The deployment of PHM systems carries substantial financial implications, prompting airlines and operators to carefully evaluate the economic investment. The task of measuring the advantages of improved maintenance, reduced downtime, and increased safety presents a considerable difficulty in determining a measurable return on investment (ROI);The ability to adapt is of utmost importance for PHM systems, as they must contain the capacity to adjust and accommodate a wide range of aircraft types and fleet sizes. The effective implementation of PHM technologies in diverse aircraft poses a significant and complex undertaking.

In order to address these challenges in a comprehensive manner, it is crucial to cultivate a spirit of collaboration among many key actors, including aircraft manufacturers, maintenance providers, regulatory entities, and technological innovators. The application of PHM in the aviation sector requires addressing specific problems, notwithstanding its promise to improve safety and efficiency.

## 2. Prognostics Approaches

There exist the following three distinct methodologies for data analysis in the context of prognostic:Physics-based models (PbM);Data-driven models (DdM);Knowledge-based models (KbM).

The physics-based methodology necessitates a precise representation of the physical system’s behavior, encompassing both its normal and defective states. The inference of a system’s health may be made by comparing the data obtained from sensors with the predictions of the model. Physical techniques encompass the utilization of PoF models. One approach to crack growth analysis involves the integration of experiments, observation, geometric analysis, and condition monitoring data to assess the extent of damage caused by a certain failure mechanism.

The data-driven methodology employs historical data on prior actions to ascertain current performance and forecast the RUL and are based on the utilization of past run-to-failure (RTF) data. These strategies are frequently employed for estimate purposes, relying on a pre-established threshold for failure. The application of wavelet packet decomposition technique and/or hidden Markov models (HMMs) can be utilized to improve the accuracy of results by incorporating time frequency data, hence providing greater precision in comparison to solely examining time variables. Nevertheless, the methodologies that utilize historical data for asset life prediction need a comprehensive understanding of the asset’s physical characteristics.

### 2.1. Physics-Based Approaches

The fundamental assumption postulates the presence of a tangible framework that clarifies the process of deterioration. As a result of this reasoning, the physical model is occasionally referred to as a degradation model, whereas the physics-based prognostics is commonly known as a model-based prognostics [21]. The main objective of this section is to present essential physics-based prognostics algorithms and analyze the challenges related to their practical application.

If there exists a precise physical model that effectively characterizes the deterioration of damage over time, then the field of prognostics may be considered substantially addressed. This is due to the fact that the future behavior of damage can be ascertained by advancing the degradation model into subsequent time periods. In practical use, it is important to note that the degradation model may not be fully comprehensive, and there is uncertainty regarding the future usage conditions. Hence, the primary concerns in physics-based prognostics pertain to enhancing the precision of deterioration models and integrating future uncertainties. For an instance, the Paris model provides an explanation for the process of fracture propagation under fatigue loading conditions, with the stress intensity factor range serving as a representative parameter. The rate at which fractures form depends on the level of stress intensity, which can vary due to different usage conditions. Additionally, the rate at which the damage will increase is determined by two model parameters, namely, m and C. The uncertainty of these characteristics arises from the variability inherent in the production process. Furthermore, it should be noted that the Paris model is specifically formulated for the analysis of an infinite plate subjected to model fatigue loading. In practical applications, it is common for plates to possess finite dimensions and be subject to boundary restrictions that impose constraints when interacting with other components. Hence, it is imperative that the decision-making process pertaining to prognostics include the many sources of uncertainty and is grounded on a cautious assessment of damage deterioration. 

The methodology of physics-based prognostics is depicted in Figure 7. The deterioration model is formulated as a mathematical function that depends on the usage (or loading) circumstances, denoted as L, as well as the elapsed time, denoted as t, and the model parameters, denoted as θ. The primary source of uncertainty is from usage conditions, which are characterized by unknown future usages; however, it is commonly believed that usage conditions and time are predetermined when developing physics-based models. Given this premise, the primary emphasis of PbM pertains to the identification of model parameters and the prediction of future deterioration patterns.

While it is feasible to obtain the model parameters through laboratory experiments, it is crucial to acknowledge that the actual model parameters utilized in a specific system may differ from those obtained through laboratory investigations. For example, when different batches of materials are used inside a specific system, they have discernible characteristics that differ from the material employed in that system. In order to accommodate the inherent variability in material properties among different batches, material handbooks frequently encompass a wide spectrum of material features.

The effectiveness of prognostics can be significantly influenced by the existence of uncertainty in model parameters. For example, it has been demonstrated that the exponent (m) for aluminum alloys in the Paris model falls within the range between 3.6 and 4.2. The phenomenon explains a modest 16% of the observed variability, although the life cycle can demonstrate a significant range of up to 500%. If a conservative estimation is performed, it is possible that maintenance may need to be scheduled around once after approximately 20% of the overall lifespan has been utilized. Therefore, it is imperative to reduce the degree of uncertainty linked to model parameters to improve the precision of forecasting the RUL and, consequently, the duration of maintenance.

Once the model parameters have been determined through the updating process, it becomes feasible to anticipate the future deterioration behavior by extending the model to future time periods. This involves substituting the identified parameters into the degradation model, along with time and loadings. The prediction of RUL is achieved by continuously propagating the deterioration condition until it surpasses a predetermined threshold.

Parameter estimation techniques serve as criteria for categorizing various physics-based methods. There are several strategies for model parameter identification, including nonlinear least squares (NLS), the Bayesian method (BM), and multiple filtering-based approaches such as the Kalman filter (KF) [22] and particle filter (PF). The filtering techniques employed are grounded in the principles of the recursive Bayesian approach. The KF provides the precise posterior distribution in the context of a linear system that is subject to Gaussian noise. To enhance the performance for nonlinear systems, other approaches within the KF family have been devised, including the extended KF (EKF) and the unscented KF (UKF) [23,24].

The paper [24] utilized three kinds of real-time model-based methods, including the EKF, UKF, and PF, to estimate the state-of-charge (SOC) of sodium-ion batteries (SIBs). In their study, the authors of [24,25] put forth a physics-informed smooth particle filter (SPF) framework aimed at predicting the RUL of lithium-ion batteries (LiBs). This framework involves the estimation of parameters associated with a single particle model (SPM) of LiBs, with a focus on identifying and extracting three primary degradation mechanisms. In another study, reference [26] employed the usage of PF to estimate wear coefficients in centrifugal pumps. They also devised a prognostics methodology based on a model, whereby the task of defining various damage development routes is approached as a joint problem. The utilization of PF is proposed to estimate the parameters associated with the damage and degradation mode of a battery degradation model. Additionally, a crack growth model is employed to elucidate the process of updating model parameters, damage progression, and prediction of RUL. This approach addresses the challenge of estimating the state parameters involved in RUL prediction [27].

The KF-family and PF algorithms, as aforementioned, employ a filtering approach that iteratively changes parameters by including individual measurement data. The efficacy of the KF family is heavily influenced by the starting condition of the parameters and the variance of the parameter, as well as the accuracy of the linearization approximation. In contrast, the utilization of PF is not subject to any limitations with regards to systems and the type of noise. 

The selection of a prognostic’s technique should be dependent on many application parameters, including but not limited to data type, uncertainty, data noise, and data size [28]. It is crucial to acknowledge that no universal algorithm can cater to the requirements of every system. Several instances of physics-based prognostics can be found in the literature, such as a battery deterioration model and a crack development model [29]. The approaches integrate a physical model with observable data to ascertain model parameters, which in turn enable the prediction of the RUL [27].

A technique is proposed for prognostics in battery deterioration utilizing a combination of a physical model and observed data to estimate model parameters. These parameters are then used to forecast the RUL of the battery [30]. The influence of model parameters on model behavior is significant and frequently not well understood, necessitating their identification as an integral component of the prognostic process [31]. Various techniques can be employed to estimate model parameters, including KF and PF [32].

In terms of integrating physics-based models in hybrid prognostics, one possible approach involves the incorporation of PbMs with DdMs, such as machine learning influenced by physics principles, to improve the capabilities of prognostics. The authors of [33] proposed a framework utilizing physics-based performance models to deduce unobservable model parameters pertaining to the health of a system’s components through the resolution of a calibration problem. The aforementioned factors are later integrated with sensor inputs and employed as inputs to a deep neural network, resulting in the creation of a data-driven prognostics model that incorporate physics-augmented characteristics. The authors of [34] utilized a modeling approach that incorporates a natural probabilistic interpretation of the prognostics exercise. A comprehensive evaluation of various classifier models is conducted on two actual datasets derived from the aeronautics industry. The findings suggest that deep learning classifier approaches are very appropriate for prognostics of this nature and have the potential to outperform traditional classification techniques by a substantial margin.

The paper [35] introduces a hybrid modeling methodology that integrates principles of physics into deep neural networks. Reduced-order models capture a significant portion of the input–output relationship; however, the utilization of data-driven kernels serves to minimize the disparity between forecasts and observations. The entire battery discharge is represented using a reduced-order model that is based on the Nernst and Butler–Volmer equations. Additionally, the battery’s non-ideal voltage is modeled using a multilayer perceptron.

The authors of [36] present a proposed multi-physics model that operates under a limited number of simplifying assumptions. The model includes a solution for the behavior of the lubricating film using the finite difference method. The model is subsequently implemented on an established, empirically verified model of the flight control actuators of a currently operational, large scale commercial aircraft in the face of excessive backlash. Developing novel health monitoring methods for detecting fault initiation and tracking the progression of rod ends till failure circumstances is realized by establishing a physics-based model of these components. As proposed by [37], the field of epigenetics offers valuable insights into the influence of environmental influences on the expression of an organism’s genes. This knowledge contributes to our understanding of the overall health of biological systems and can be utilized to make predictions about their future states. The relationship between environmental influences and epigenetic alterations, which subsequently gives rise to visible features, can be associated with conditions that impact the overall health of a system.

The work of [38] introduces a novel approach rooted in physics-based principles, referred to as a model order reduction (MOR) method, to simulate the dynamics of aircraft. The concept of employing a physics-based learning approach involves the integration of the fundamental principles of aircraft dynamical systems into machine learning models, with the aim of minimizing training expenses and improving simulation capabilities. The research indicates that the physics-based learning approach demonstrates enhanced computational efficiency in comparison to a traditional numerical method. This is due to the capacity of the physics-based learning method to employ larger time step sizes, which violate the numerical stability constraint, while maintaining an explicit time integration scheme.

The present work of [39] introduces a physics-based model to elucidate the phenomenon of blockage. The suggested model is based on a well-established pressure drop equation and possesses the capacity to mimic three distinct stages of the clogging phenomena. The proposed model incorporates particle filters to create predictions pertaining to future levels of clogging and to estimate the remaining lifespan of fuel filters. The results indicate that the approach utilized in this research produces prognostic forecasts that exhibit a high level of accuracy and precision.

Physics-informed machine learning (PIML) is an approach that integrates data with the fundamental laws of physics, enabling the utilization of models that may include incomplete physical knowledge in a coherent manner. The representation of this phenomena may be effectively achieved by employing automated differentiation and neural networks that are especially designed to create predictions that conform to the fundamental principles of physics [40]. The incorporation of PIML allows for the alignment of the model with physical principles, hence enabling the steering of the model towards appropriate solutions. As a result, the utilization of PIML leads to an improvement in both the precision and the effectiveness of the model, especially in scenarios that are marked by unpredictability and a large number of variables [40,41].

#### 2.1.1. Challenges in Physics-Based Prognostics

In contrast to data-driven methodologies in next section, physics-based prognostics algorithms provide several advantages. Physics-based approaches have the capability to create long-term predictions. Once the model parameters have been reliably discovered, it becomes feasible to forecast the RUL by propagating the physical model until deterioration approaches a preset threshold. Additionally, physics-based methodologies need a very little amount of data. In a theoretical context, it is conceivable to ascertain the parameters of a model when the quantity of available data is equivalent to the number of unknown parameters inside the model. In practice, however, a larger quantity of data is necessary due to the presence of noise in the data and the insensitivity of degradation behavior to parameters. It is worth noting that physics-based prognostics algorithms often require a smaller amount of data compared to data-driven techniques.

There are the following three significant concerns in the realm of physics-based prognostics that hinder its practicality: model suitability, estimation of parameters, and data sources. 

##### Model Suitability

The problem of model adequacy pertains to the extent to which the physical model possesses the capability to accurately forecast the future deterioration behavior. The issue of curve-fitting in regression differs slightly from the typical approach, since regression focuses on the accuracy between data points, which may be seen as the error in the interpolating area. In the field of prognostics, there is a particular focus on the analysis of several data points, specifically the mistake associated with extrapolation. Physics-based models are advantageous in forecasting the long-term behaviors of damage due to their utilization of a physical model that describes the behavior of damage. Prior to any further analysis, it is imperative to conduct model validation as a first step, as the majority of models inherently involve assumptions and approximations. There has been a significant research institution dedicated to the validation of models using statistical methods, including hypothesis testing and Bayesian approaches [42,43].

In general, when the level of intricacy in a model increase, there is a corresponding increase in the quantity of model parameters. Consequently, the estimation of these parameters becomes more arduous. The authors in references [44,45] have shown that addressing the issue of model adequacy can be alleviated by identifying the comparable parameters from a simpler model. The prediction of crack formation in complex geometries was achieved through the utilization of a simplified Paris model, in which an assumed stress intensity component was employed. The model parameters were modified to accommodate the inherent imprecision linked to the stress intensity factor. The focus of this paper is limited to the comparison of damage behavior between basic and elaborate models. This approach eliminates the need for additional validation techniques to assure the accuracy of the models.

##### Estimation of Parameters

The process of parameter estimation has significant importance in physics-based prognostics since it enables the uncomplicated prediction of RUL after the model parameters have been known. Physics-based parameter estimation involves addressing two distinct difficulties. 

The estimation accuracy is influenced by the properties of different approaches;The presence of correlations between model parameters, as well as between model parameters and loading circumstances, poses challenges in accurately identifying the parameters.

An efficient prognostics algorithm demonstrates the capacity to accurately estimate model parameters with a little amount of data. Although there may be difficulties in properly identifying components, it is still possible to obtain precise predictions in deterioration and RUL.

##### The Dataset Pertaining to the Source of Failure

The utilization of structural health monitoring (SHM) data is commonly employed to forecast and anticipate the model parameters of a system that is in operation. The data acquired from SHM may demonstrate a noteworthy level of noise and bias due to several factors, including the specific attributes of the sensor equipment employed and the prevailing conditions within the measurement environment. Noise is the term used to describe the random fluctuations that are noticed in measured data or signals. These fluctuations occur due to the interference or unwanted electromagnetic fields present in electronic equipment. Bias is a phenomenon characterized by the constant deviation of signals from their true values, sometimes caused by calibration problems. The existence of noise presents difficulties in effectively distinguishing signals linked to degradation, while bias causes mistakes in prediction results. The investigation of noise mitigation and bias correction has become popular within the prognostics discipline.

Reference [46] proposed a prognostics methodology for identifying the decline in performance of multilayer ceramic capacitors when subjected to temperature–humidity–bias circumstances. Reference [47] proposed a technique for developing interpretive prognostics for switch mode power supplies with electromagnetic input filters. This approach involves modeling the degradation trajectories of sub-components and utilizing discrete event simulation to generate lifecycle data related to the system impedances. These data are then employed as inputs for machine learning-based prognostics, enabling the generation of interpretable predictions regarding the RUL of the system. The presence of noise in sensor signals is a significant obstacle to the accurate identification of deterioration features. This interference has a detrimental impact on the prognostics capabilities of both physics-based and data-driven systems. In signal processing, the process of mitigating this issue is often accomplished by the application of de-noising techniques. The utilization of a multilevel hierarchical kriging (MHK) model was suggested by [48] to expedite the convergence of a high-fidelity aero structural optimization of helicopter rotors towards the global optimum. This model has the capability to include three or more degrees of fidelity. 

### 2.2. Data-Driven Models

Data-driven models involve the acquisition of knowledge regarding the behavior of a system through the analysis and interpretation of data obtained from different engineering systems. These approaches strive to employ a neutral and implicit approach to the learn the system by leveraging raw data gathered from real-world observations. Researchers investigate the correlations among multiple variables and observations, revealing unforeseen patterns in the natural world. This method allows for the identification of new scientific concepts and, in certain instances, enables predictions to be made even in the absence of existing experiences [49].

The behavioral systems theory defines a system as a set of trajectories, which are patterns of behavior. As a result, this definition possesses an innate inclination towards approaches that depend on empirical data for the purpose of analysis and research. The determination of system representations, input/output partitioning of variables, zero beginning state, and other assumptions, is not predetermined. The data-driven methodology enables a perspective of a dynamical system that is devoid of any specific representation, instead viewing it as a compilation of trajectories [50]. The application of algorithms to identify patterns within data and make predictions about new data is fundamental to data-driven approaches in classification. 

Data-driven modeling refers to the application of empirical measures and machine learning techniques to efficiently develop models that can predict and prevent issues before they occur. The complexity of the models is smaller in comparison to physics-based models; however, their dependability may also be reduced [51]. To improve accuracy and reliability, the integration of data-driven models with other methodologies, such as physics-based modeling and feature selection strategies, has been suggested [52,53].

The implementation of a data-driven strategy involves a series of consecutive stages. Several pieces of advice aimed at facilitating the modeling process are suggested in Figure 8 as follows:

A wide array of data analysis tools exists, encompassing both rudimentary spreadsheet applications and sophisticated business intelligence systems. Several commonly used tools for data analysis include the following:Python v3.11 is a widely utilized programming language that is well regarded in the fields of data analysis and machine learning;R v4.3 is a computer language and software environment that is specifically designed for statistical computation and graphics;SAS v8 is a comprehensive software package that encompasses a wide range of applications, including advanced analytics, multivariate analysis, business intelligence, data management, and predictive analytics;Excel is a spreadsheet application that is extensively utilized and provides a range of features for data analysis and visualization;Power BI is a business analytics service developed by Microsoft that offers a range of interactive visualizations and business intelligence functionalities;Tableau v2023.2 is a software tool designed for data visualization, enabling users to obtain insights and comprehension from their data;Apache Spark is an open-source distributed computing system designed for the purpose of processing large-scale data;MATLAB R2023b is a well-known software that has well-developed built-in toolboxes and applications that can be useful to researchers working on data analytics;JMP Pro 17 is powerful statistical software designed with scientists and engineers solving problems with data, which is packed with tools for data preparation, analysis, graphing.

The aforementioned software provides a diverse array of functionalities for anyone involved in data analysis and encompasses tasks such as data cleansing, manipulation, visualization, and modeling. The selection of an appropriate tool is of utmost significance, taking into consideration both the specific requirements and the proficiency level of the user. 

The theoretical understanding of data analysis design involves the procedure of formulating the problem as an endeavor to acquire knowledge from the data, considering an unknown input distribution. Researchers are currently engaged in the development of robust computational and statistical techniques for the design of combinatorial algorithms driven by data. This pertains to both offline and online scenarios, wherein a set of representative problem instances from a specific application are presented either simultaneously or sequentially, respectively [54]. The aims of algorithm development grounded on data analysis closely correspond to those of algorithms that hold the potential to improve their own performance. The main finding suggests that it is feasible to utilize and enhance approaches established from learning theory to achieve these goals in various algorithmic contexts [55].

#### 2.2.1. Neural Networks (NNs) and Learning Methods

Neural networks (NNs) are derived from computational models that emulate the structural organization of the human brain. Artificial neural networks (ANNs), originally conceptualized by McCulloch and Pitts in 1943, are computational models that aim to emulate the cognitive processes of the human brain involved in learning. Complex systems are simulated and analyzed using known input/output instances. In contrast to the neural structures seen in the human brain, the algorithm under consideration operates by means of a network of interconnected pre-processing units, facilitating the processing of data. The NN can be regarded as an opaque entity. The black box, known as NN, generates a set of proposed actions to solve a given scheduling instance, producing results that cannot be derived from a known mathematical function [56].

An NN is a dynamic system consisting of several artificial neurons that are coupled to create a sophisticated network. These neurons possess the ability to modify their structure in response to internal or external input. Put simply, this model is not explicitly designed to address a certain problem. Instead, it acquires the ability to tackle such problems through a training or learning procedure that involves the use of instances. The dataset referred to as the training set consists of input data paired with their respective output values. The approach closely replicates the cognitive capacity of the human brain to acquire knowledge from past encounters.

Numerous algorithms are employed in the training of neural networks, exhibiting a wide range of variations. Several often-used terms in the field include feedforward, backpropagation, gradient descent, cost function, and sigmoid. Figure 9 depicts a simplified feedforward ANN framework.

During the conclusion of a forward pass in the training phase, the output layer receives the predictions (network outputs) H from the preceding layer and computes the loss O by comparing these predictions with the training objectives. The output layer calculates the partial derivatives of the loss function O with respect to the predicted values H and transmits (propagates) these results to the preceding layer. Figure 10 illustrates the sequential movement of data inside a convolutional neural network, culminating at the output layer.

The inaccuracy of NNs is determined during a testing phase, when the network’s predictive capability is assessed while altering the weights of its connections. Once a training set of examples has been constructed using historical data and the appropriate architecture, such as feedforward networks or recurrent networks, the subsequent crucial stage in the implementation of NNs is the learning process. During the training process, the NN can deduce the connections between the input and output, establishing the relative weights of the connections between individual neurons. Each neuron computes a weighted sum of its inputs and generates a binary signal if the cumulative input surpasses a specific activation threshold. As a result of this mechanism, the network can successfully execute highly intricate tasks. Learning algorithms may be classified into several types [57].

A conventional NN consists of artificial neurons, also known as units, organized in a hierarchical structure of interconnected layers, where each layer is connected to the next levels. The quantity of units might vary significantly, ranging from a few hundred to several million units. Certain units, referred to as input units, are specifically built to receive diverse types of information from the external environment, which the network will endeavor to acquire knowledge, identify, or otherwise analyze. The output units are situated on the opposing side of the network and are responsible for indicating their response to the acquired information. The artificial brain consists of layers of hidden units positioned between the input units and output units, collectively constituting most of its structure. 

The majority of NNs exhibit complete connectivity, wherein every hidden unit and output unit establishes connections with all units in the next layers. The associations between individual units are denoted by a numerical value known as a weight, which can assume either a positive value or a negative value. The greater the weight, the greater the impact that one unit has on another. This phenomenon is analogous to the intercellular communication observed in the human brain, where neurons stimulate each other through synaptic junctions [58].

The transmission of information inside an NN occurs through two distinct pathways. During the process of learning or subsequent operation following training, the NN receives patterns of information through the input units. These input units then activate the hidden units, which then propagate signals to the output units. The architecture that is commonly referred to as a feedforward network is known by this name. Not all units are in a state of constant operation. In NN, every unit is subject to receiving inputs from the units positioned to its left. These inputs are thereafter subjected to multiplication by the weights associated with the connections via which they traverse. Each unit in a network accumulates the inputs it receives and, in the case of the simplest sort of network, if the sum exceeds a specific threshold value, the unit becomes activated and subsequently activates the units it is linked to [59].

In order for an NN to acquire knowledge, it is necessary to incorporate an analogous feedback mechanism to how toddlers learn via receiving guidance on their correct or incorrect actions. Feedback is a ubiquitous tool employed by individuals for the purpose of learning on a regular basis. 

NNs acquire knowledge by a feedback mechanism known as backpropagation, which is the standard method employed for learning. This process entails the comparison of the network’s generated output with its intended output, and subsequently utilizing the disparity between the two to adjust the weights of the connections between the units within the network. This adjustment process occurs in a reverse manner, starting from the output units, passing through the hidden units, and concluding at the input units. Over time, the backpropagation algorithm facilitates the learning process of the NN by minimizing the discrepancy between the observed output and the desired output, ultimately leading to a state where the two align perfectly. Consequently, the network achieves optimal performance by accurately comprehending the underlying patterns and relationships [59].

The primary task entails converting information into a meaningful output. NNs can exhibit either feedforward or feedback behavior, determined by the direction in which information is propagated [60,61].

Feedforward networks are a type of NN architecture in which signals propagate in a unidirectional manner, moving from the input layer towards the output layer. These NNs consist of a solitary input layer and a solitary output layer, with the possibility of containing several hidden layers or none at all. The information flow may be divided into the following two distinct stages: the learning phase, which occurs when the network is being taught, and the regular operating phase, which takes place after the network has completed its training process. Feedforward networks are commonly employed in the field of pattern recognition;Feedback networks, particularly recurrent or interactive networks, utilize memory, known as their internal state, to effectively process input sequences. Network loops facilitate the transmission of signals in both directions. Feedback networks are frequently utilized within the framework of time series or sequential processes.

Learning algorithms are classified into the following types [62]:

Supervised learning involves the acquisition of knowledge by a network through the analysis of known instances derived from past data, enabling the network to establish connections between input and output;Weak supervision, alternatively known as semi-supervised learning, is a prominent approach within the field of machine learning. Its significance and prominence have been amplified in recent times, particularly with the emergence of extensive language models. This is primarily attributed to the substantial volume of data necessary for effectively training these models. The approach is distinguished by its use of a limited quantity of human-labeled data, which is solely employed in the more resource-intensive and time-consuming supervised learning framework. This is then followed by the utilization of a substantial quantity of unlabeled data, which is exclusively employed in the unsupervised learning framework. Put simply, the desired output values are only given for a portion of the training data;Unsupervised learning refers to a type of learning where just the input values are provided, without any explicit labels or guidance. In this context, it is observed that comparable stimulations tend to activate neurons that are near each other, whereas dissimilar stimulations tend to activate neurons that are further apart.

Supervised learning is a machine learning paradigm that leverages annotated datasets to facilitate the training of algorithms with the objective of accurately classifying data or predicting outcomes. Supervised learning involves the provision of an algorithm with a collection of input–output pairs, with the objective of acquiring a comprehensive rule that establishes a mapping between the given inputs and outputs. Supervised learning is extensively employed in various domains, including but not limited to speech recognition, mechanical engineering, and aerospace. An instance of supervised learning can be observed in the context of generalized methodology for the PM of aircraft systems. The paper of [63] is exemplified through its utilization of three distinct test cases, namely, the engine, the environmental control system, and the fuel system. It provides a comprehensive description of the digital twin configuration, simulation parameters for both normal and problematic scenarios, and a diagnosis approach based on OSA-CBM as previously depicted in Figure 1. The process of diagnostics is conducted sequentially for each system, employing four supervised machine learning algorithms. The most effective method for each system will thereafter be utilized in a vehicle-level reasoner known as FAVER (A Framework for Aerospace Vehicle Reasoning). This reasoner relies on these system diagnoses as an initial reference for vehicle reasoning and the resolution of fault ambiguity.

In contrast, semi-supervised learning refers to a machine learning approach that involves the integration of a limited quantity of labeled data with a substantial quantity of unlabeled data during the training process. Semi-supervised learning occupies an intermediate position between unsupervised learning, which lacks labeled training data, and supervised learning, which relies solely on labeled training data. The objective of semi-supervised learning is to leverage unlabeled data to enhance the efficacy of a model that has been trained on labelled data. In this study, reference [64] provides an innovative semi-supervised prognostic model designed for systems that exhibit partially observable failure modes. Specifically, our model addresses situations where the training dataset contains only a limited number of systems with known failure modes. Initially, a graph-based semi-supervised learning approach is devised to extract distinctive properties that delineate the various failure scenarios. Subsequently, the variables, along with the multi-sensor streams, are utilized as inputs for an elastic net functional regression model to forecast the RUL.

Unsupervised learning algorithms are utilized for the analysis of data, enabling the grouping into distinct segments according to the shared characteristics or disparities. An illustration can be found within the domain of developing health indicators and RUL prognostics. The authors of [65] utilize unsupervised learning techniques for the development of health indicators in systems characterized by a number of failures. The autoencoder is trained using unlabeled data samples so that the actual RUL is unknown. The autoencoder incorporates the diverse operating characteristics of aircraft, such as variable altitude and speed, into its framework. The health indicators are subsequently employed to forecast the RUL of the aircraft system through the utilization of a similarity-based matching methodology. The methodology employed in the study yields precise RUL estimations, exhibiting a root mean square error (RMSE) of merely 2.67 flights.

The efficacy of machine learning models, regardless of whether they fall under the categories of supervised, unsupervised, or semi-supervised learning, is contingent upon various factors. These factors encompass the caliber and volume of the data employed for model training, the selection of algorithm, and the intricacy of the problem. In general, supervised learning algorithms are often observed to exhibit higher accuracy compared to unsupervised learning algorithms due to their utilization of labeled data for model training. This enables the algorithm to acquire knowledge from established input–output pairs and enhance the precision of its predictions. Nevertheless, the efficacy of supervised learning algorithms may be constrained by the accessibility and caliber of annotated data.

Unsupervised learning algorithms abstain from utilizing labeled data and instead rely on identifying inherent patterns and correlations within the data. As a result, the accuracy of their results may vary depending on the complexity of the task and the quality of the data. Semi-supervised learning algorithms use elements from both supervised and unsupervised learning methodologies. The utilization of a restricted amount of annotated data is employed to augment the effectiveness of the model that has been trained on unannotated data. The effectiveness of semi-supervised learning algorithms has the potential to exceed that of unsupervised learning algorithms; however, this outcome is dependent on the availability and quality of annotated data. In summary, the accuracy of machine learning models depends on various parameters, and there is no generally applicable method to find the most precise type of machine learning. Table 2 presents more details regarding supervised and unsupervised learning methodologies.

The utilization of the backpropagation algorithm in supervised learning has emerged as the predominant approach for training neural networks [66]. 

#### 2.2.2. Recurrent Neural Network (RNN) and Long Short-Term Memory (LSTM)

LSTM is an abbreviation for long short-term memory. This particular neural network variant finds application in the domains of artificial intelligence and deep learning. In contrast to conventional feedforward neural networks, LSTM networks possess feedback connections, hence classifying them as a variant of recurrent neural network (RNN). The utilization of LSTM enables the processing of not just individual data points, such as images, but also full sequences of data.

Within the field of aviation prognosis, LSTM networks have demonstrated considerable efficacy in the analysis and prediction of aircraft trajectory data. A proposed model for trajectory prediction is an attention-based LSTM model, comprising two separate components. During the initial phase, the LSTM model is utilized to extract the temporal features of the flight trajectory. In the following section, the attention mechanism is employed to effectively manage the processed sequence features. The attention mechanism operates by amplifying the significance of primary aspects while reducing the importance of minor elements [67,68].

A RNN is a specific variant of an artificial neural network (ANN) that allows for connections between nodes. This unique characteristic enables the output of nodes to impact the future input of those same nodes. This feature allows it to exhibit temporal dynamic behavior. RNNs can use their internal state, referred to as memory, to proficiently comprehend input sequences that possess diverse lengths. These characteristics make them appropriate for various applications, including unsegmented, linked handwriting recognition or speech recognition. RNNs are distinguished by their capacity to integrate information from preceding inputs to modify the current input and output. Conventional deep neural networks operate on the assumption that the input and output variables are mutually independent. In the context of RNNs, it is important to note that the output is influenced by the preceding parts within the sequence.

Reference [69] conducts a comprehensive analysis and assessment of different prognostic models for aviation engines’ RUL and seeks to compare the effectiveness of these models with an LSTM technique, which utilizes a data-driven machine learning approach. This paper utilizes the C-MAPSS datasets to assess the performance and outcomes of each technique. The results obtained indicate that the utilization of the modified LSTM technique incorporating an attention mechanism yields enhanced predictive accuracy for the RUL estimation of aviation engines, hence exhibiting superior performance. Another study introduces a novel approach that employs an LSTM network, a specialized architecture intended for identifying concealed patterns in time series data. The objective of this approach is to monitor system degradation and estimate the exhaust gas temperature (EGT). The effectiveness of the suggested methodology is assessed by employing health monitoring data related to turbofan engines used in aircraft. The network’s ability to recognize the input data as a real-time sequence enables the possibility of predicting the output in the following stage. The results of the study indicate a significant ability to forecast the outcome in the following period. Additionally, the model being examined demonstrates a diminished rate of learning over time and improved precision [70].

#### 2.2.3. K-Means

The k-means clustering algorithm is a type of unsupervised learning technique that aims to divide a given dataset into k distinct groups, taking into consideration the similarity and distance metrics between data points. Every individual data point is assigned to the cluster that has the closest mean value, which acts as a representative prototype for that cluster. The k-means clustering algorithm aims to minimize the total sum of squared distances between the data points and the cluster centroids. The centroids of the clusters are determined by calculating the arithmetic mean of the data points within each cluster. The method of k-means clustering involves iteratively reassigning data points to clusters until a convergence condition is satisfied.

As an illustration, research introduces a novel hybrid data preparation model with the aim of enhancing the accuracy of failure count prediction. The suggested approach operates in two distinct stages. In the initial phase, the ReliefF technique, which is a feature selection approach employed for attribute evaluation, is utilized to identify the most impactful and least impactful factors. During the second step, the k-means algorithm undergoes modifications to effectively remove noisy or inconsistent data points. The evaluation of the hybrid data preparation model’s performance is conducted on the maintenance dataset pertaining to the equipment [71]. The study of [72] proposes a novel degradation prognostics strategy for aeroengines operating under various conditions. A k-means algorithm and three defined indicators are combined to distinguish between various operational conditions, then monotonic and trending degradation features are extracted. A deep forest classifier (DFC) and an LSTM are employed to develop an offline health state estimation model and a degradation trend prediction model. The results demonstrate that the proposed strategy for aeroengines in a variety of operating conditions is effective and realizable.

Multiple techniques exist for determining the optimal number of clusters in k-means clustering. One approach that can be used is the elbow method, which entails calculating the sum of squared distances (referred to as the within-cluster sum of squares) for various values of k and selecting the value of k at which the drop in the sum of squared distances starts to stabilize. Silhouette analysis is an additional technique that can be employed to examine the degree of separation between groups generated by a clustering algorithm. It also enables the evaluation of parameters, such as the optimal number of clusters. One example of the utilization of k-means clustering in the field of aircraft systems involves the detection and categorization of malfunctions in quadcopter unmanned aerial vehicles (UAVs). The objective of the study outlined in reference [73] was to design a failure detection system for UAVs by analyzing vibration data and employing the k-means clustering algorithm. The purpose was to accurately identify any potential issues that may arise during UAV flights. The results demonstrated that the combination of the gyroscope parameter in the vertical direction and the accelerometer parameter in the same direction resulted in the most accurate detection of failures during emergency landings of malfunctioning UAVs.

#### 2.2.4. Density-Based Spatial Clustering of Applications with Noise (DBSCAN) Algorithm

In addition to the k-means method, a number of clustering methods are available. One example that can be used to illustrate this concept is the density-based spatial clustering of applications with noise (DBSCAN) algorithm. DBSCAN is a data clustering algorithm proposed by Martin Ester, Hans-Peter Kriegel, Jörg Sander and Xiaowei Xu in 1996 [74]. The DBSCAN algorithm is a density-based clustering technique that can identify clusters of different shapes and sizes, as well as identify outlier points that do not belong to any one cluster. The DBSCAN algorithm functions by defining a local neighborhood around each individual data point and subsequently finding and grouping points that are near form clusters. The points that do not fall into any cluster are classified as noise. The approach exhibits two main parameters, specifically the radius of the neighborhood encompassing each data point and the minimum number of points required to form a dense region.

DBSCAN exhibits several advantages in comparison to k-means, encompassing its capability to identify clusters of diverse shapes and sizes, as well as its proficiency in managing noise and outliers. Nevertheless, the algorithm’s performance can be influenced by the selection of parameters and may exhibit suboptimal results when applied to datasets with significant variations in density.

The methodology involves the creation of a limited area around each individual data point, followed by the grouping together of points that are near separate clusters. The points that do not fall into any cluster are classified as noise. The algorithm exhibits two main parameters, specifically the ɛ parameter, which signifies the radius of the neighborhood encompassing each data point, and the ‘minPts’ parameter which indicates the minimum number of core points required to establish a dense region.

The process initiates by choosing a point at random and retrieving all points that fall within its ɛ neighborhood. A new cluster is formed if the number of points in the neighborhood is equal to, or more than the minimum number of points required. The algorithm proceeds to extend the cluster by including all points within the ɛ neighborhood of each point in the cluster. If the count of points within the vicinity is below the minimum threshold ‘minPts’, the point is designated as noise. 

Following is an illustration of how to perform DBSCAN on input data. The DBSCAN algorithm can cluster a 2-D circular dataset using the Euclidean distance metric as the default. Using the squared Euclidean distance metric, it is also essential to compare the results of the DBSCAN and k-means clustering algorithms applied to the dataset. Figure 11 depicts the existence of two distinct clusters within the dataset with different metrics. The first step is to generate a dataset consisting of two noisy circular patterns. Specify values ɛ = 1, while minPts = 5. DBSCAN accurately divides the dataset into two clusters using the Euclidean distance metric. Then, employ the DBSCAN clustering algorithm in conjunction with the squared Euclidean distance metric. Alternately, specify ɛ equal to 1 and minPts equal to 5. The application of the squared Euclidean distance metric within the DBSCAN algorithm detects and distinguishes the two clusters present in the given dataset with precision. The implementation of k-means clustering using the squared Euclidean distance metric follows. Specify the value k for the number of clusters, which is 2. The application of the squared Euclidean distance metric in k-means clustering leads to an inaccurate identification of the two clusters present in the provided dataset. A novel enhancement to the DBSCAN method is suggested in [75], whereby dynamic time warping (DTW) is used to effectively tackle the clustering concern. The suggested method is validated using two distinct flight datasets derived from fleet data, which exhibit varying lengths. The experimental findings demonstrate that the enhanced DBSCAN algorithm has the capability to identify potential anomalous flying behaviors during both the ascent and descent phases. In [76], several deep neural network autoencoder architectures were trained on nominal data to calculate a health indicator using the anomaly scores. The findings indicate a positive correlation between high anomaly scores and the presence of detected failures in the maintenance logs. Additionally, many scenarios exhibit a rise in the anomaly score for multiple flights preceding the breakdown of the system, hence providing a viable approach for early detection of faults.

#### 2.2.5. Fuzzy Logic

The recognition of the importance of inference in the management of uncertainty is growing within engineering applications. In situations where engineers and scientists encounter numerical difficulties that cannot be effectively addressed by standard mathematical methods, the utilization of fuzzy logic is frequently considered as a feasible solution. Fuzzy logic facilitates the characterization and control of a system that lacks a formally recognized or precisely defined model [77]. The field of fuzzy theory offers the capacity to effectively represent and simulate human cognitive processes related to common sense reasoning and decision-making.

Fuzzy logic is an extension of Boolean logic based on the mathematical principles of fuzzy sets, which provide a broader interpretation of the classical set theory. Incorporating the notion of degree into condition verification, fuzzy logic offers significant flexibility in reasoning. This permits conditions to exist in states beyond true and false. Therefore, fuzzy logic permits the consideration of errors and uncertainties during decision-making processes. There is a tenuous connection between hazy logic and probability theory. Commonly, the Bayesian framework is used to characterize probability approaches that deal with imprecise knowledge [78,79]. It is essential to observe that fuzzy logic does not always necessitate a probabilistic justification. The generalization of the findings pertinent to multivalued logic with the intention of retaining a portion of the underlying algebraic structure is a common method.

The study of [80] introduces a methodology for forecasting the RUL of a generic system, specifically focusing on achieving a greater level of interpretability in the prediction model. The utilization of established computational intelligence methodologies, including decision trees, fuzzy logic, and genetic algorithms, facilitates the development of a composite framework known as a genetic fuzzy rule-based system (GFRBS), which involves the automatic generation of fuzzy rules and the subsequent tuning of the membership functions that relate to these rules. The proposed methodology is implemented in a case study examining the deterioration of aircraft engines. Fuzzy logic is currently used in a number of industrial and consumer electronics devices that require an effective control system but where optimal control is not necessarily a concern [81]. 

#### 2.2.6. Decision Trees

A decision tree is a decision assistance tool that uses a visual representation or model in the form of a tree to illustrate various options and their corresponding consequences. This tool is employed to determine the outcome of occurrences that are characterized by uncertainty. One strategy for presenting an algorithm that consists solely of conditional control statements is by utilizing a specific methodology. A decision tree is a visual depiction that has a resemblance to a flowchart. The structure is comprised of internal nodes that serve as attribute tests, such as evaluating the outcome of a coin flip as either heads or tails. The branches of the tree symbolize the potential outcomes of the tests, while the leaf nodes symbolize the class labels, which are the decisions made after evaluating all qualities. The paths that span from the root to the leaf nodes serve as representations of categorization criteria.

Tree-based learning algorithms are commonly acknowledged as highly powerful and often used methodologies for supervised learning. Predictive models possess the inherent capability to improve the precision, consistency, and comprehensibility of outcomes. These models effectively capture the complexities of nonlinear interactions, enabling the analysis of various types of issues such as classification and regression tasks. There are several frequently used terms that are linked with decision trees [82].

The root node functions as the central representation of the complete sample or population and is subsequently divided into two or more subsets that exhibit comparable characteristics;The notion of splitting entails the process of dividing a certain node into multiple sub-nodes;A decision node can be defined as a sub-node that bifurcates into other sub-nodes;Leaf or terminal nodes are defined as nodes that do not experience any additional splitting;Pruning refers to the removal of sub-nodes from a decision node. The opposite of splitting is the act of combining or merging;In the context of tree structures, a branch or sub-tree denotes a specific component or subdivision inside the larger hierarchical framework;The concept of a parent node and child node is a fundamental aspect in the field of computer science and data structures. In a hierarchical structure, such as a tree, a parent node is defined as a node that has one or more child nodes directly connected. In the context of a hierarchical structure, a node that is partitioned into smaller nodes is referred to as a parent node, while the smaller nodes are considered as child nodes.

The decision tree is a supervised learning technique that is employed in classification situations when there exists a predetermined target variable. This approach is applicable to both categorical and continuous input and output variables. The sample is divided into two or more homogenous subsets, often known as sub-populations, depending on the most significant splitter or differentiator found in the input variables [82]. Decision trees have an inherent structure characterized by a “if…then…else” framework, rendering them very compatible with programmed structures. Moreover, they exhibit a high degree of suitability for classification tasks when traits or features are methodically examined to ascertain a definitive category. 

An instance of employing a decision tree is in aircraft predictive maintenance applications of airplane [83]. In the paper, the hybrid algorithm’s architecture comprises a decision tree, wherein each node represents a neural network that has been trained to perform binary classification for a particular output category. It demonstrates the ability to categorize data of diverse volume and variety accurately and efficiently. This exemplifies the algorithm’s practicality in real-life situations, while also highlighting the advantages of integrating decision trees and neural networks rather than utilizing them separately. Another research article [84] presents a new approach for predicting the RUL of aircraft engines using a deep bidirectional recurrent neural networks (DBRNNs) ensemble method. The iterative training process of multiple regression decision tree (RDT) models involves changing the weights of components in the domain. The experimental findings demonstrate that the proposed methodology has attained a higher level of performance in comparison to other established approaches.

#### 2.2.7. Support Vector Machine (SVM)

Support Vector Machines (SVMs) are a type of supervised learning algorithms that are commonly employed for tasks such as classification, regression, and outlier identification. SVMs are typically employed in the context of classification tasks. Support vectors are the specific coordinates of individual observations. In the SVM method, every data item is represented as a point in an n-dimensional space, where n is the number of features. The numerical representation of each feature corresponds to the value of a certain coordinate [85]. As depicted in Table 3, SVMs have a number of advantages and disadvantages.

Artificial neural networks, support vector machines, decision trees, and k-means clustering are distinct machine learning techniques used for classification tasks, and each is distinguished by its own operational mechanisms. Table 4 summarizes how machine learning methods function.

The process of selecting an appropriate machine learning model for a given dataset can be intricate. When making a model selection, it is important to consider many aspects. The aspects encompassed within this particular context encompass the dimensions and properties of the dataset, the intricacy of the problem being addressed, as well as the performance, interpretability, and sustainability of the model. One method for model selection involves running trials using several algorithms and later evaluating their performance on the given dataset. Techniques like cross-validation may be employed to evaluate the effectiveness of each model in making predictions on unseen data. The careful evaluation of trade-offs between the performance of a model and its other properties, such as explainability and complexity, has significant significance.

In addition to assessing the effectiveness of the model, it is advisable to analyze the clarity of the results. Several algorithms, such linear regression and decision trees, exhibit a reasonably straightforward interpretability, while others, such as neural networks, pose more significant hurdles in terms of interpretability. The consideration of model complexity is an additional factor that requires careful attention. Sophisticated models provide the capability to detect intricate patterns within data; however, this advantage comes at the cost of heightened challenges in terms of maintenance and interpretability. The size of the dataset should be carefully taken into account, as certain algorithms demonstrate better performance when used to big datasets, while others are more proficient in handling smaller datasets. The choice of the most suitable data model is contingent upon the specific needs and objectives at hand. The subsequent paragraphs outline the overall benefits and drawbacks linked to the utilization of these algorithms in prognostic applications. 

Table 5 presents a comprehensive overview of the benefits and drawbacks linked to different techniques. 

#### 2.2.8. Anomaly Detection Algorithms

Anomaly detection is a technique used to detect and identify patterns that deviate from expected behaviors, also known as outliers. There are numerous applications of artificial intelligence (AI). These applications range from detecting potential intrusion attempts by identifying unusual patterns in network traffic to monitoring the health of systems by identifying malignant tumors in magnetic resonance imaging scans. In addition, AI is used to identify faults in operating environments and aerospace engineering [86].

Similarities exist between anomaly detection, noise reduction, and novelty detection; however, it is not identical to these ideas. Novelty detection is concerned with identifying previously unobserved patterns among newly observed data points that were not included in the original training dataset. Noise reduction refers to the process of minimizing the influence of unwanted observations on the analysis, thereby removing noise from a signal with inherent significance [86]. There are three major sorts of methods for detecting anomalies.

Density-based anomaly identification is a method employed for the purpose of detecting anomalies or outliers within a given dataset. This methodology relies on analyzing the density distribution of the data points to identify and isolate such abnormalities;Clustering-based anomaly detection is an approach employed to find anomalies in each dataset through the utilization of clustering methodologies;SVM-based anomaly detection refers to the utilization of SVMs for the purpose of identifying anomalies within a given dataset.

##### Density-Based Anomaly

The concept of density-based anomaly identification refers to a method used in data analysis to identify anomalies or outliers in a dataset based on the density distribution of the data points.

Density-based anomaly detection relies on the utilization of the k-Nearest Neighbors (kNN) technique. The underlying premise posits that typical data points are concentrated within proximity, whereas anomalies are situated at a considerable distance. The proximity of a given collection of data points is assessed using a scoring mechanism, such as the Euclidean distance or a comparable metric, which is contingent upon the nature of the data being analyzed, whether it is categorical or numerical in nature. The kNN algorithm is a straightforward and non-parametric approach for passive learning. It is commonly employed for data classification by using distance metrics like Euclidean, Manhattan, Minkowski, or Hamming distance to identify similarities between data points. Data can also be categorized according to its relative density [87]. This phenomenon is commonly referred to as the local outlier factor. The foundation of this idea is rooted in a distance metric known as reachability distance.

##### Clustering-Based Anomaly

Clustering is well recognized as a prominent idea within the field of unsupervised learning. The fundamental premise posits that data points exhibiting similarity are inclined to be members of comparable groupings or clusters, as ascertained by their proximity to local centroids. The k-means technique is extensively employed in the field of clustering. The algorithm generates k groups of data points that exhibit similarity. Anomalies may be identified for data instances that do not belong to these groupings [88].

##### Anomaly Detection with SVM

As previously mentioned, SVMs are well recognized as a highly efficient approach for the identification of anomalies and are frequently utilized in supervised learning scenarios. Nevertheless, some adaptations, such as the implementation of core vector machine (CVM) [89], facilitate the detection of irregularities in an unsupervised fashion, even in scenarios where the training dataset does not possess any labeled data. The method utilizes the training set to acquire knowledge of a flexible border to categorize the normal data samples. Following this, the system adapts its parameters according to the specific testing instance to identify any anomalies that fall beyond the established range. 

The results generated by an anomaly detection system can exhibit variability, contingent upon the specific application or context in which it is employed. The system has the capability to generate numerical scalar values that can be utilized for data filtration according to domain-specific criteria, as well as textual labels that offer descriptive information.

#### 2.2.9. Conventional Numerical Techniques

##### Kalman Filter (KF)

The Kalman filter (KF), initially proposed by Kalman in 1960, has been extensively utilized in various practical fields, notably in the realms of aeronautics and aerospace. With the growing number of applications, some concerns have been recognized, one of which is the issue of divergence. The issue at hand is a result of the inherent lack of reliability in the numerical methodology employed or the inaccurate portrayal of the system under investigation [90]. The KF is a method that leverages a series of measurements observed over a duration, incorporating statistical noise and other sources of flaws, to provide predictions of unknown variables that demonstrate enhanced accuracy in comparison to estimations generated solely from a solitary observation. The operation of the system incorporates a two-phase mechanism consisting of prediction and update stages. During the prediction phase, the KF generates estimates of the current state variables, together with their accompanying uncertainties. Following the observation of the subsequent measurement outcome, the estimates are modified using a weighted average method, wherein estimates with higher levels of certainty are accorded greater emphasis [91].

The KF is extensively employed in various technical disciplines. One commonly observed use is to the guidance, navigation, and control of many types of vehicles, particularly airplanes, satellites, and dynamically positioned ships. Furthermore, the application of the KF is widespread in the realm of time series analysis, namely, in disciplines such as signal processing and econometrics. The KF is a widely recognized topic in the domain of robotic motion planning and control, and it is also employed in trajectory optimization. 

The square root filter (SRF) has been suggested as a more reliable alternative to the Kalman filter (KF) [92]. It is expected that incorporating numerically stable orthogonal transformations at each iteration will improve the precision of the filter estimations. The utilization of Sequential Monte Carlo Recursive Filtering (SMCRF) has a higher computational burden in comparison to the conventional KF. As a result, researchers have developed alternate versions of the SRF, such as UDU-algorithms and the Chandrasekhar form [93]. The efficiency of these implementations can be enhanced to achieve or even exceed that of the standard KF, especially when considering the Chandrasekhar SRF, within appropriate experimental circumstances [94].

Since its initial use in a variety of applications, the numerical stability issues associated with KF have been well-recognized. The optimality of the estimation procedure implies a susceptibility to various forms of errors. Commonly employed strategies to address these stability concerns include the following [93]:One potential enhancement would be to increase the level of mathematical precision;One such approach is to use a square root filtering technique;The covariance matrix should be symmetrized at each stage;It is recommended to initialize the covariance appropriately in order to mitigate significant fluctuations;One such approach is to employ a fading memory filter;Utilize hypothetical process noise.

The investigation of improving the execution speed of the KF has not been thoroughly examined. To enhance robustness and enable the integration of fixed-point arithmetic, hence enabling the migration of a design to high-speed digital signal processors, it is imperative for the covariance matrix to exhibit the characteristics of symmetry and positive definiteness. If these criteria are not met, the covariance matrix cannot effectively reflect statistical information related to the components of the state vector. During the early stages of KF applications, it was recognized that factored form SRFs were the preferred option for developing applications requiring a significant degree of operational reliability. The elements of the covariance matrix are decomposed and subsequently propagated forward during the measurement process, undergoing updates at each measurement step. The positive semi-definiteness of the covariance matrix can be demonstrated by expressing it as the result of multiplying its constituent elements.

The ${UDU}^{T}$ filter is a commonly employed factored-form KF [95]. The covariance matrix P may be expressed in terms of the matrix elements U and D in Equation (1) as follows.
(1)$$P={UDU}^{T}$$

The ${UDU}^{T}$ KF formulation has exhibited effective performance and dedicated digital implementations have been developed to provide comparable processing speeds for factored covariance filters and conventional covariance propagation methods.

The components of the covariance matrix can be regarded as in Equation (2):(2)$$P_{ij}=\sigma_{i}\sigma_{j}\rho_{ij}$$

The symbol $P_{ij}$ represents ${ij}^{th}$ element of the covariance matrix. The symbol $\sigma_{i}$ is the standard deviation of the estimate for the $i^{th}$ component of the state, whereas $\rho_{ij}$ represents the correlation coefficient between the $i^{th}$ and $j^{th}$ components of the state. Both $\sigma_{i}$ and $\rho_{ij}$ encompass significant physical information that characterizes the advancement of KF estimation. This information pertains to the present efficacy of estimate and the potential occurrence of numerical challenges in the future. Nevertheless, the individual components included in matrices utilized for factored-form filters lack meaningful physical significance until the covariance matrix is calculated, along with the statistical parameters in Equation (2).

The optimality of the KF is dependent on the assumption that the model is accurate, which makes it intolerant of any model errors. The values of the initial state may be completely unknown, necessitating the use of approximations for the initial states along with significant standard deviations. In addition, starting state correlations are frequently unknown and assumed to be zero. Frequently, the use of convenient beginning conditions results in the occurrence of extremely pronounced first transient phenomena and premature filter failure.

Recognizing that instability frequently arises because of acquiring an excessive quantity of information quickly during the estimation process is one strategy that could be employed. To avoid potential di-convergence, it is preferable to have a delayed rate of convergence and/or a lower threshold for the predicted standard deviation. Calculating the updated covariance using the physics-based model parameters and evaluating the resulting covariance matrix is one possible method for achieving this result. If a substantial improvement in estimation is anticipated, it is possible to increase the levels of measurement noise or process noise and then recalculate the covariance update prior to continuing data processing at this juncture. Utilizing an iterative process yields a constrained information filter in which the potential increase in standard deviation for each state is re-restricted. Similarly, it is possible to impose a restriction on the lower limit of a state’s standard deviation by specifying either an absolute number or a percentage of the initial level of uncertainty. The primary difficulties associated with adaptive techniques are primarily computational in nature. A filter form can be designed rapidly to iteratively execute covariance update computations, thereby modifying noise models to limit the extraction of information.

The difficulties associated early KF implementations have been alleviated as a result of the increased computational cost. Notwithstanding, KF solutions are rarely considered for high-speed embedded applications [96]. One of the limitations associated with the fundamental KF is its underlying assumption of linearity. Therefore, numerous altered versions of the KF have been proposed to circumvent these numerical challenges. Several of these changes are derived from heuristics, such as the stabilized KF or the ordinary KF with reduced bounds. Consequently, the successful implementation of these modifications typically necessitates a greater level of knowledge.

The presence of nonlinearity can be attributed to either the process model, the observation model, or both. The Extended Kalman Filter (EKF) and Unscented Kalman Filter (UKF) are the prevailing forms of KFs utilized for nonlinear systems [97].

##### Extended Kalman Filter (EKF)

The Extended Kalman Filter (EKF) is a variant of the KF that is intended for nonlinear systems. This is achieved by linearizing the system dynamics around a current estimate of the mean and covariance. In the fields of nonlinear state estimation, navigation systems, and the Global Positioning System (GPS), the EKF is widely regarded as the established standard when dealing with clearly characterized transition models [98]. The EKF addresses nonlinearity by linearizing state transition and observation models around an estimate of the current mean and covariance. The state transition and observation models are not required to manifest linearity with respect to state variables within the EKF framework. These models may alternatively be represented by differentiable functions. At each time step, the current anticipated states are used to build and evaluate the Jacobian matrix of partial derivatives. The use of these matrices is applicable within the KF’s equations. Effectively transforming the nonlinear function into a linear approximation in the vicinity of the current estimate.

In the EKF, the Jacobian is used to linearize the state transition and observation models around an approximation of the current mean and covariance. Throughout each time iteration, the Jacobian matrix, which comprises of partial derivatives, is calculated, and evaluated using the current anticipated states. Utilizing these matrices within the KF equations facilitates the nearly linear propagation of the state and state covariance. 

The EKF state propagation procedure consists of the following two discrete phases: prediction and update. During the prediction stage, the current state is estimated by utilizing a series of previous state estimations. As it is based on previous projections and lacks empirical evidence regarding the current state of the system, the projected estimate may be regarded as prior knowledge. During the update procedure, the prior estimate and the current data are combined to generate an estimate of the current and future states of the system. Typically, these processes are performed iteratively in an alternating fashion, with prediction occurring until the next observation, followed by an update process utilizing the most recent observations. 

To optimize the efficacy of an EKF, its parameters must be modified. The process noise, which describes the degree of variation or ambiguity between the actual movement of the object and the selected motion model, is an essential parameter that can be modified. By modulating the process noise, the filter can assign greater significance to more recent observations than to older ones. This allows the filter to accommodate changes in direction or velocity. A further important parameter that can be modified is the measurement noise, which characterizes the level of measurement uncertainty. By manipulating the level of measurement noise, the filter can allocate greater weight to readings that are deemed more trustworthy. 

The authors of [99] present a novel approach for prognostics, which integrates the EKF with a newly designed linearization technique. The prognostics approach being proposed has been formulated within the specific context of fatigue fracture propagation in fuselage panels. In this scenario, the model parameters are not known, and the crack propagation is subject to various forms of uncertainty. The findings indicate that the coupled EKF-linearization approach yields favorable outcomes. Specifically, the EKF algorithm effectively identifies the model parameters, and the linearization method produces prediction results that are comparable to those obtained by the Monte Carlo method; moreover, this approach offers substantial computational savings. Another research article [100] presents an approach for optimizing predictive line maintenance of redundant aeronautical equipment under various wear situations. The estimation of degradation patterns and future wear values is conducted by employing a multiple model approach of EKF technique. The effectiveness and value of the proposed methodology are demonstrated by a case study that utilizes field prognostics data from hydraulic systems.

##### Particle Filter (PF)

Particle filters (PFs), also known as the sequential Monte Carlo method, are able to effectively address problems with substantial nonlinearity without the need for linearization. The concept underlying the PF can be summed up as follows. Consider a circumstance in which the mathematical model represents a nonlinear stochastic dynamic system. The goal is to estimate the hidden states of the system by integrating model predictions with imperfect and insufficient observations of the system. The Bayes filter, also known as the optimal filter, can be used for this purpose [101]. Using Bayes’ rule, the posterior PDF of the concealed states is estimated iteratively. The absence of a closed-form analytical expression for the posterior distribution in most cases presents a challenge. Numerous approximation techniques, including the EKF, have been proposed as potential solutions to this problem. In cases where the system exhibits excessive nonlinearity or the posterior distribution deviates substantially from Gaussian characteristics, the technique may encounter difficulties [102]. By utilizing Monte Carlo sampling, the PF method is used to estimate the posterior distribution. This method eliminates the need to presume linearity in the dynamic model or Gaussian noise distribution. 

Particles, which are isolated random realizations sampled directly from the state space, are utilized by PF. These particles are used to represent the posterior probability and to facilitate the revision of the posterior by incorporating new observations. Utilizing the Bayesian formula, the particle system is precisely positioned, assigned weights, and propagated recursively.

Since its inception, PF algorithm has found applications in several domains, including but not limited to signal processing, economics, robotics, and geophysics.

The paper of [103] introduces a methodology for adaptive data-driven prognostics reasoning. The prognostic reasoning approach has been demonstrated through the utilization of a case study on the turbofan jet engine in the field of engineering. This article uses the NN to construct the nominal model, while the PF is utilized to monitor the current degradation and degradation parameters. The study conducted by [104] demonstrates the utilization of the PF methodology in the context of forecasting the degradation of steam generator tubes. Authors can effectively showcase the range of prediction outcomes by employing a case study, hence highlighting the influence of uncertainty levels linked to measurement data. The paper by [105] introduces a robust particle filtering approach that systematically forecasts future health conditions and creates a probability distribution for the expected health state based on the number of selected particles. Insufficient attention has been given to the study of errors arising from the numerical implementation of the particle filter.

In practice, the state equations in nearly all systems necessitate numerical solutions, which unavoidably introduce errors into the filtering process [106,107]. The RUL of a hydraulic pump was estimated using an adaptive-order particle filter. The prediction was accomplished by the utilization of a state recognition method that depended on wavelet packet norm entropy. This method involved monitoring the deteriorating trend of the pump. Significantly, this methodology was utilized within complex operational environments [2].

Another example can be found in a research study that investigated the process of determining the mass and propulsion settings of departing aircraft using a recursive particle filter. The methodology employed in this research was based on a nonlinear state-space system derived from aircraft point-mass performance models, as cited in reference [108]. The investigation observed the deterioration pattern of the hydraulic pump by employing a state identification technique that relies on wavelet packet norm entropy, particularly in the presence of intricate operational circumstances. According to the findings, the adaptive-order PF demonstrated effectiveness in forecasting the RUL of the hydraulic pump. 

In addition to the applications related to the use of PFs for RUL, there are further applications of PFs in the prognostication of aviation systems. A research investigation was undertaken to assess the effectiveness of four advanced PFs of online crack development prognosis utilizing guide wave-based SHM. The enhanced PFs considered were the auxiliary particle filter (APF), regularized particle filter (RPF), dual regularized particle filter (DRPF), and guide wave-based marginalized particle filter (GW-MPPF). The study utilized a fatigue test on attachment lugs to substantiate its findings. Additionally, the study analyzed two distinct scenarios, with a specific focus on assessing the precision of the measurement equation [108].

##### Regression

Regression analysis is a statistical method used to estimate an overdetermined system, which is defined as a system with more equations than unknown variables. This method demonstrates its benefits in situations where the identification of a precise solution becomes too complex as a result of measurement mistakes or random noise present in the dataset. The utilization of it demonstrates a broad spectrum of practical implementations across several academic disciplines [109]. The utilization of this tool allows for the assessment of the extent of the relationship between the variables, as well as the capacity to develop prediction models for their future interactions. Regression analysis consists of the following three primary types: linear regression, multiple linear regression, and nonlinear regression. The linear and multilinear models are commonly acknowledged as the predominant models in the field. Nonlinear regression models are commonly used in the examination of intricate datasets that exhibit nonlinear associations between the dependent and independent variables [109]. Figure 12 depicts the three classifications.

Linear regression is widely recognized as the most used regression model in academic and practical contexts. In this model, the objective is to forecast the outcome of n data points $\left(x_{1},y_{1}\right),\;\left(x_{2},y_{2}\right),\;\ldots ,\;\left(x_{n},y_{n}\right)$ using a regression model expressed as in Equation (3):(3)$$y=a_{0}+a_{1}x$$
where $a_{0}$ and $a_{1}$ represent the fixed parameters of the regression model.

One way to assess the quality of the prediction made by the linear model $a_{0}+a_{1}x$ for the response variable y is by examining the amount of the residual $\varepsilon_{i}$ at each of the n data points. The equation may be expressed as follows:(4)$$E_{i}=y_{i}-(a_{0}+a_{1}x)$$

In an optimal situation, if all the residual $\varepsilon_{i}$ are equal to zero, it is conceivable that we have identified an equation in which all data points adhere to the model. Hence, the objective of estimating the regression coefficients is to minimize the residual. The least squares method is widely acknowledged as the predominant methodology employed to minimize the residual. The approach employed in this methodology involves the selection of the model’s constants estimates in a manner that aims to minimize the sum of the squared residuals, as indicated in the reference [110]; that is $\sum_{i=1}^{n}E_{i}^{2}$.

Nonlinear regression is a statistical technique employed to describe the connections present in observational data by utilizing a nonlinear combination of the model parameters and one or more independent variables. Certain nonlinear regression problems have the potential to be converted into the linear domain [110,111].

In practical applications, researchers often start the analysis by picking a suitable model for estimation. Subsequently, they employ their preferred methodology, such as ordinary least squares, to estimate the parameters of the chosen model. Regression models often consist of several components [112]. The parameters that are not known are often represented as a scalar or vector β. The independent variables, often represented as a vector $X_{i}$, are observed within the dataset. The index i is used to indicate a specific row of data. The dependent variable, commonly represented as the scalar $Y_{i}$, is the observed variable in the dataset. The error terms, commonly represented as the scalar $e_{i}$, are unobservable in the collected data.

It should be noted that in many fields of application, other terminology is employed in lieu of dependent and independent variables. Many regression models posit that the dependent variable $Y_{i}$ may be expressed as a function of the independent variable $X_{i}$ and the parameter β, as in Equation (5):(5)$$Y_{i}=f\left(X_{i},\beta \right)+e_{i}$$The variable $e_{i}$ in this context denotes an error term that accounts for unmodeled factors influencing $Y_{i}$ or random statistical noise.

The objective is to determine the function $f\left(X_{i},\;\beta \right)$ that provides the best possible fit to the given data. The functional form of this function may be derived from a priori knowledge of the relationships between $Y_{i}$ and $X_{i}$, without relying on the specific facts at hand. In the absence of accessible knowledge, the analyst opts for a form that is adaptable. As an illustration, the utilization of a basic univariate regression model in Equation (6):(6)$$f\left(X_{i},\beta \right)=\beta_{0}+\beta_{1}X_{i}$$
which implies that the Equation (7):(7)$$Y_{i}=\beta_{0}+\beta_{1}X_{i}+e_{i}$$
provides a reasonable approximation of the underlying statistical process responsible for creating the observed data.

After a statistical model is confirmed, there are several techniques at disposal to estimate the parameters β. For instance, the least squares technique, including ordinary least squares, aims to determine the optimal value of β by minimizing the sum of squared errors. The regression technique that is used will ultimately yield an estimation of the parameter β, often represented as $\sum_{i}{\left(Y_{i}-f(X_{i},\beta )\right)}^{2}$, to differentiate it from the actual (unknown) parameter value that produced the data. 

The utilization of least squares is prevalent due to its ability to provide an estimated function $f(X_{i},\hat{\beta })$ that closely approximates the conditional expectation $E(X_{i}|Y_{i})$. However, alternative methods such as least absolute deviations or quantile regression can be employed to effectively describe other functions $f\left(X_{i},\;\beta \right)$.

Adequate data are required to make an accurate estimation of a regression model. If N observations are processed, each consists of a dependent variable and two independent variables $\left(Y_{i},\;X_{1i},\;X_{2i}\right)$. Additionally, considering the scenario where to estimate a bivariate linear model as in Equation (8):(8)$$Y_{i}=\beta_{0}+\beta_{1}X_{1i}+\beta_{2}X_{2i}+e_{i}$$
using the method of least squares. When provided with a limited number of N=2 data points, there are an endless number of combinations $\left(\hat{\beta },{\hat{\beta }}_{1},{\hat{\beta }}_{2}\right)$ that can equally explain the observed data. Any combination that fulfills the given condition
(9)$$\hat{Y_{i}}=\hat{\beta_{0}}+\hat{\beta_{1}}X_{1i}+\hat{\beta_{2}}X_{2i}$$
can be selected. All these factors contribute to and, thus, represent legitimate solutions that minimize the sum of squared residuals in Equation (10)
(10)$$\sum_{i}{\hat{e}}_{i}^{2}=\sum_{i}({\hat{Y}}_{i}-({\hat{\beta }}_{0}+{\hat{\beta }}_{1}X_{1i}+{\hat{\beta }}_{2}X_{2i}){)}^{2}=0$$

The presence of a multitude of solutions can be ascribed to the underdetermined characteristic of the equation system N=2, which encompasses three variables of uncertain values. In an alternative methodology, one can utilize visualization tools to depict a multitude of three-dimensional planes that cross at a constant quantity of two points, designated as N=2.

To do an estimation of a least squares model with k unique parameters, it is necessary to ensure that the number of different data points N is equal to or greater than k. In cases when N exceeds k, it is typically not possible to identify a collection of parameters that can precisely capture the data. The amount N−k is a common occurrence in regression analysis, where $\left(X_{1i},\;X_{2i},\ldots ,\;X_{ki}\right)$ represents a set of variables. It is essential for these variables to be linearly independent, meaning that it is not possible to rebuild any individual variable by combining or multiplying the other ones. This condition guarantees that the matrix $X^{T}X$ is invertible, hence implying the existence of a solution β [113].

#### 2.2.10. Statistical Approaches

##### Gamma Process

A gamma process is a stochastic process characterized by the property of having increments that are independently distributed according to the gamma distribution [114]. The process, denoted as Γ(t, γ, λ), is commonly represented as a pure jump rising Lévy process. It possesses an intensity measure $v\left(x\right)=\gamma x^{-1}\exp(-\lambda x)$ for x greater than zero. Jumps with magnitudes falling within the range [x, x+dx) are seen to follow a Poisson process characterized by an intensity function $v\left(x\right)dx$. The parameter γ governs the rate at which jump intervals occur, whereas the scaling parameter γ  determines the inverse relationship between jump size and γ. The process is postulated to commence with an initial value of 0 at time t=0.

The gamma process may be characterized by its parameters, namely, the mean (μ) and variance (υ) of the increase per unit time. These parameters are related to the shape (γ) and rate (λ) parameters of the gamma distribution, where γ is equal to the square of the mean divided by the variance $\gamma =\mu^{2}/\upsilon$ and λ  is equal to the mean divided by the variance λ=μ/υ. One possible strategy for mitigating the difficulties arising from the scarcity of data and inherent uncertainties is to utilize a stochastic process model to perform time-dependent reliability evaluations of structures. The utilization of the stochastic gamma process model is deemed appropriate for the representation of the unidirectional progression of a deterioration process, such as corrosion. The gamma process is a stochastic process that exhibits independent and non-positive increments. The increments in question adhere to a gamma distribution, wherein the scale parameter remains constant, and the shape parameter fluctuates with time.

The integration of temporal uncertainty in the evolution of degradation is achieved by the utilization of a stochastic process model. The gamma process is a suitable selection for modeling the gradual accrual of deterioration over a period, particularly in scenarios related to phenomena such as wear, fatigue, corrosion, fracture propagation, erosion, consumption, creep, swell, and the deterioration of a health indicator. One advantage of employing gamma processes for the purpose of simulating degradation processes is in the inherent simplicity of the required mathematical computations [115].

##### Hidden Markov Model (HMM)

The application of a Markov chain is beneficial in scenarios where it becomes essential to compute the probability linked to a sequence of observable events. In other cases, however, certain occurrences that capture the attention remain hidden, as they are not observable. In most cases, the inclusion is not commonly observed within a given context. The elements are occasionally denoted as ‘hidden tags’ owing to their absence of direct observation [116]. The utilization of a hidden Markov model (HMM) enables the incorporation of both observable events and latent events, which are regarded as causal components within the probabilistic framework. The components that define an HMM are outlined in Table 6 [117,118]. 

Let $X_{t}$ and $Y_{t}$ be continuous-time stochastic processes. The pair ($X_{t}$, $Y_{t}$) is a HMM if:

The state of the process $X_{n}$ (resp. $X_{t}$) are called hidden states, and $P\left(Y_{n}\in A|X_{n}=x_{n}\right)$ (resp. $P\left(Y_{t}\in A|X_{t}=B_{t}\right)$) is called emission probability or output probability.

In the context of academic concern, a first order refers to the initial level or primary stage of a certain phenomenon or concept. HMMs are based on the instantiation of two simple assumptions. Similar to a first order Markov chain, the probability of a certain state is contingent solely upon the preceding state. The Markov assumption is a fundamental concept in probability theory and stochastic processes. The conditional probability of $q_{i}$ given $q_{1}\ldots q_{i-1}$ is denoted as in Equation (11):(11)$$P\left(q_{i}|q_{1}\ldots q_{i-1}\right)=P(q_{i}|q_{i-1})$$Furthermore, it is important to note that the likelihood of an output observation $q_{i}$ is solely determined by the state that generated the observation and is not influenced by any other states or observations [117] in Equation (12):(12)$$P\left(o_{i}|q_{1}\ldots q_{i},\ldots ,q_{T},o_{1}\ldots o_{i},\ldots ,o_{T}\right)=P(o_{i}|q_{i})$$

##### Relevance Vector Machine (SVM)

The relevance vector machine (RVM) approach is based on a solid theoretical framework rooted in probability theory, particularly in the comprehension of Bayes’ theorem and Gaussian distributions. This entails a thorough understanding of marginal and conditional Gaussian distributions. In addition, it assumes a prerequisite level of proficiency in matrix differentiation, the application of vector representation in regression analysis, and the comprehension of kernel functions [119]. The RVM is a machine learning approach that employs Bayesian inference to obtain succinct answers for regression and probabilistic classification tasks. The functional architecture of the RVM bears a striking resemblance to that of the SVM, albeit with a noteworthy differentiation in its ability to provide probabilistic classification skills. The model can be regarded as a Gaussian process model with a covariance function [120,121] as in Equation (13):(13)$$k(x,x^{\prime })=\sum_{j=1}^{N}\frac{1}{a_{j}}\varphi (x,x_{j})\varphi (x^{\prime },x_{j})$$In the above expression, φ represents the kernel function, often assumed to be Gaussian. $\alpha_{j}$ denotes the variances of the prior distribution on the weight vector ω, which follows a normal distribution as $\omega \sim N(0,\;\alpha^{-1}I)$. Lastly, $X_{1},\ldots ,\;X_{N}$ represent the input vectors of the training set.

The Bayesian formulation of RVM circumvents the need for a set of free parameters that are often required in SVM, which often necessitates post-optimizations based on cross-validation. Nevertheless, RVMs employ an expectation-maximization (EM) learning approach, which renders them susceptible to encountering local minima. In contrast, the techniques used by SVMs, namely, the typical sequential minimum optimization-based algorithms, have been proven to ensure the discovery of a global optimum [122,123].

##### Autoregressive Model (AR)

Autoregressive (AR) models, also known as conditional models, Markov models, or transition models, are employed to predict future behavior through the analysis of past behavior. Time series forecasting is utilized in situations where there is a discernible relationship between the values within a time series and the preceding and subsequent values. The technique basically entails conducting a linear regression analysis on the data contained within the given series, with the objective of establishing a correlation between one or more past values within the same series.

The AR model posits a direct association between the value of the outcome variable Y at a specific time point t, and the predictor variable X. The phenomenon under consideration bears resemblance to the one observed in basic linear regression models. One key differentiation between standard linear regression and AR models is in the inherent characteristics of the dependent variable Y. In AR models, the dependent variable Y is influenced not only by the independent variable X, but also by preceding values of Y.

Autoregression is a stochastic process distinguished by the presence of intrinsic uncertainty or unpredictability. Owing to the inherent probabilistic character of occurrences, the prediction can be reasonably dependable, albeit unlikely, to attain complete precision with a 100% degree of accuracy. Usually, the information that is presented is deemed to be adequately precise, hence enabling its use across a diverse array of scenarios [124,125].

AR models are utilized in the disciplines of statistics, econometrics, and signal processing to provide mathematical structures for describing a certain category of random processes. As a result, it is utilized for the examination and depiction of dynamic events observed in several fields, including the natural sciences and economics. The AR model posits that the dependent variable is impacted by a linear mixture of its previous values along with a stochastic component, which accounts for inherent unpredictability. As a result, the model can be mathematically represented as a stochastic difference equation, distinct from a differential equation. The utilization of the exponential smoothing model, in combination with the moving average (MA) model, is a specific and integral component of the broader autoregressive moving average model (ARMA) and AR-integrated MA (ARIMA) models employed in the analysis of time series data. These models are known for their complex stochastic frameworks. Furthermore, this can be seen as a specific case of the vector autoregressive (VAR) model, which covers a collection of interrelated stochastic difference equations involving several evolving random variables. In contrast to the MA model, the AR model may not demonstrate stationarity as a result of the potential inclusion of a unit root [124,126].

The paper [127] presents a fused ensemble learning algorithm integrated with various methods, such as random forests (RFs), RNN), AR model, RVM, and others to enhance predictive performance. The experimental findings have demonstrated that the ensemble learning algorithm has a notable level of robustness in predicting the RUL of aviation engines. Another paper [128] introduces a novel approach to prognostics modeling, which utilizes a nonlinear autoregressive neural network (NARNET) to estimate the RUL of a deteriorating system in the presence of dynamic operating conditions. The paper demonstrates that the incorporation of an operating condition forecast into the prognostics model yields improved accuracy and efficiency in predicting the RUL.

##### Multivariate Adaptive Regression Splines (MARS)

The multivariate adaptive regression splines (MARS) model is a non-parametric modeling strategy that improves upon linear models by incorporating nonlinearities and interactions among variables, which is a multifunctional method that enhances the efficiency of developing prediction models. These tasks encompass activities like finding relevant elements, altering independent variables, addressing missing data, and employing a self-test to mitigate the potential problem of overfitting. Additionally, it exhibits the capacity to generate predictions by considering structural factors that may influence the dependent variable, thereby creating hypothetical models. The potential of the outcome is such that it has the capacity to establish important thresholds within a collection of data sequences [129].

The construction of a proficient regression model requires a considerable amount of time and a high level of competence in the field of modeling. However, the use of MARS allows for the systematic and automated creation of regression models, thus removing the limitations imposed by the assumptions that traditional regression models must adhere to. In conclusion, this particular methodology possesses the capacity to reveal complex patterns and relationships that may present difficulties, if not impossibilities, for alternative methodologies to determine [130].

The MARS technique can be considered as an extension of recursive partitioning regression. It involves the partitioning of the predictor variable space into several subregions. The model can be expressed in mathematical notation as in Equation (14):(14)$$y_{t}=f(x_{t})=\beta_{0}+\sum_{i=1}^{k}\beta_{i}\beta (x_{it})$$

The response variable at time t is denoted as $y_{t}$, whereas the model parameters for the corresponding variables $x_{it}$, where i ranges from 1 to k, are represented by $\beta_{i}$. The parameter $\beta_{0}$ denotes the intercept, which corresponds to the value of the dependent variable when all independent variables are equal to zero. Additionally, the term “base functions” refers to the fundamental functions used as a basis for constructing more complex functions or models. The functions $B\left(x_{it}\right)$ are dependent on the variables $x_{it}$, with each $B\left(x_{it}\right)$ being defined as either $B\left(x_{it}\right)=\max(0,\;x_{it}-c)$ or $B\left(x_{it}\right)=\max(0,\;{c-x}_{it})$. Here, c represents a threshold value, and k denotes the number of explanatory variables, including interactions of the predictor variables. The space partition points, and mode parameters are derived from the analyzed data. The complexity of the model may be inferred from the number of generated basic functions.

The MARS method generates cut points for many variables. Points are demarcated by fundamental functions that indicate the beginning and end of a specific domain. In each partitioned portion, the fundamental function of a variable is altered to demonstrate linearity. The ultimate model is created by combining the derived fundamental functions.

To determine the cut spots, use a stepwise forward/backward stepwise method. The forward stepwise technique first generates a model that tends to overstate the true model due to the inclusion of a significant number of base functions. Following this, the reverse stepwise approach is utilized to remove the nodes that have the least influence on the total modification. The method’s termination condition is triggered when the approximation generated includes a predetermined maximum number of functions, as indicated by researchers [130]. The subsequent methods might be employed to ascertain the optimal model: Cross-validation criteria are a metric utilized in academic research to assess the performance and generalization ability of a machine learning model. The measure of fit to data and penalty refers to the evaluation of how well a particular model or hypothesis corresponds to the observed data, taking into account any costs or penalties associated with the model’s complexity or simplicity. The necessity arises due to the model’s complexity and the resultant increase in variance. Based on this criterion, it may be preferable to select a simpler model over a more complex one [131];The determination coefficient, often known as adjusted R^2^, is a statistical measure used to assess the goodness of fit of a regression model. The coefficient of determination is used to assess the appropriateness of a model by comparing the observed values with the projected values;The mean absolute error ratio is calculated based on the observed data and the projected value. The optimal model is characterized by the minimal error rate. The mean absolute ratio is commonly denoted as in Equation (15):
(15)$$\sum_{i=1}^{n}\frac{\left\{\left[\frac{Observed\_Value-\mathrm{Pr}edicted\_value}{Observed\_value}\right]\right\}}{n}$$

Various statistical software programs, such as R v4.3, MATLAB R2023b, and Python v3.11, offer the capability to implement MARS.

#### 2.2.11. Classification, Cluster Analysis, and Bayesian Techniques 

##### Classification

The classification issue is a prominent sub-field within the study of discriminant analysis and has strong ties to other statistical disciplines. Classification is the process of designating an element to the appropriate population within a given set of known populations, taking observable factors into account. Significant emphasis is placed on the development of multivariate statistics, which has wide-ranging applicability in a variety of disciplines [132]. Both theoretical and applied statisticians find this topic to be intriguing. There exist the following four primary approaches for addressing the categorization problem:

Let D be the set of observed data objects, denoted as $D=\left\{X_{1},\ldots ,\;X_{m}\right\}$. Each instance or item $X_{i}$ is represented as an ordered vector of attribute values, $X_{i}=\left\{X_{i1},\ldots ,\;X_{ik}\right\}$. The objective of classification is to identify the optimal class description h from a given space H, which can effectively predict the data D. The word ‘best’ may be understood as the hypothesis with the highest likelihood given the seen data D and some previous knowledge about the hypothesis H in the absence of D. This refers to the prior probability of the different hypotheses in H when no data have been observed. Bayes’ theorem offers a method for calculating the probability of the most optimal hypothesis, based on the prior probabilities, the probabilities of viewing the data under different hypotheses, and the actual data [17]. 

The prior probability, denoted as P(h), represents the likelihood of a hypothesis h being true prior to the observation of any evidence. Similarly, let P(D) represent the prior probability of observing the data, indicating the likelihood of D without any knowledge regarding which hypothesis is true. The notation $P\left(D|h\right)$ represents the probability of seeing the event D in a particular universe where the hypothesis h is assumed to be true. The primary challenge in unsupervised classification is to determine the probability $P\left(h|D\right)$, which represents the likelihood of hypothesis h being valid, based on the observed data D. The posterior probability of hypothesis h, denoted as $P\left(h|D\right)$, is a term used in Bayesian statistics to quantify the level of trust in hypothesis h after observing the relevant data. Consequently, the collection of data elements influences the posterior probability, although the prior probability remains unaffected by the specific dataset. Bayes’ theorem offers a computational approach for determining the posterior probability [133] like Equation (16):(16)$$P\left(h|D\right)=\frac{P\left(D|h\right)P\left(h\right)}{P\left(D\right)}$$

Or
(17)$$P\left(h|D\right)=\frac{P\left(D|h\right)P\left(h\right)}{\sum_{h}P\left(D|h\right)P\left(h\right)}$$

The theorem of total probability states that if occurrences $h_{1},\ldots ,\;h_{n}$ are mutually exclusive with $\sum_{i=1}^{n}P(h_{i})=1$, then the probability of an event may be calculated by summing the probabilities of $\sum_{i=1}^{n}P(D|h_{i})P(h_{i})=1$ given each of the mutually exclusive events $h_{1},\ldots ,\;h_{n}$, weighted by their respective probabilities.

When the set of potential values for h is continuous, the prior distribution is transformed into a differential distribution, and the summations over h are replaced by integrals; hence, the execution of this algorithm becomes challenging.

The expectation-maximization (EM) technique is a Bayesian approach employed to compute the local maximum likelihood or maximum a posteriori (MAP) estimates of parameters in situations where the model relies on unobserved latent variables. In contrast to the exhaustive consideration of all potential states of the world and corresponding hypotheses in traditional approaches, the EM method focuses on a limited set of models. This method operates on the assumption S that one of these models accurately represents the world. A model is composed of two distinct sets of parameters. The first set, denoted as T, encompasses discrete parameters that define the functional characteristics of the model. These parameters determine attributes such as the number of classes and the correlation between attributes. The second set, denoted as V, comprises the continuous parameters that assign specific values to the variables present in T. These values are necessary to fully specify the general structure of the model.

The EM algorithm aims to identify the optimal combination of variables V and T that can effectively classify a given dataset D. This is achieved by maximizing the probability of the joint distribution $P\left(DVT|S\right)$, which may be expressed as the product of the conditional probability $P\left(D|VTS\right)$ and the prior probability P(VT|S) [133,134].

##### Clustering

The issue of clustering holds considerable significance in the realm of unsupervised learning. The commonly utilized clustering methods include the subsequent approaches [135]:K-means clustering, and mixture modeling are the most prevalent approaches utilized for canonical and flat clustering;Hierarchical clustering is a data analysis technique that involves the examination of a hierarchical structure, often depicted as a tree, to discern significant patterns within the data at different levels of granularity. In contrast to alternative clustering techniques, the primary goal of this method does not include achieving a singular partition of the dataset. There exist two distinct categories of hierarchical clustering algorithms, specifically agglomerative and divisive. The initial methodology investigates techniques for consolidating individual data points with the aim of establishing a hierarchical framework. Conversely, the subsequent methodology entails the iterative partitioning of the data into progressively smaller clusters;Spectral clustering entails the calculation of a similarity matrix for every possible combination of data points. The procedure entails performing an eigenvalue decomposition, thereafter, projecting the data points onto a subspace delineated by a certain set of eigenvectors. Following this, one of the clustering methodologies previously mentioned, such as k-means or hierarchical clustering, is utilized to categorize the data into distinct clusters.

##### Bayesian Methods

Bayesian methodologies provide a systematic framework for rationalizing and making logical inferences about the surrounding environment, especially in the presence of inherent ambiguity. The Bayesian technique is based on a rigorous approach to handling uncertainty, which was initially developed by Bayes and Laplace in the 18th century and further refined by statisticians and philosophers in the 20th century. Bayesian techniques have become prominent models of human cognitive processes in various domains, such as multi-sensory integration, motor learning, visual illusions, and brain computing. Moreover, they function as the fundamental basis for machine learning systems [136].

Bayes’ theorem, sometimes known as Bayes’ rule, is a fundamental principle in probability theory. It expresses the conditional probability of an event θ given the occurrence of event x in Equation (18):(18)$$P\left(\theta |x\right)=\frac{P\left(x|\theta \right)P(\theta )}{P(x)}$$

This theorem may be derived from basic principles of probability theory. In this context, x represents a specific data point, whereas θ denotes the model parameters. The probability of θ, denoted as P(θ), is commonly known as the prior. It signifies the likelihood of θ prior to acquiring any knowledge on x. The probability of x given θ, denoted as P(x|θ), represents the likelihood in academic discourse. The posterior probability of θ, denoted as P(θ|x), is the likelihood of θ given the observation x. The normalizing constant, P(x), is a term that ensures the posterior probability distribution is properly normalized.

Given the joint probability of x and θ as P(x,θ), by marginalizing θ, thus the function P(x) may be expressed as the integral of P(x,θ) with respect to θ in Equation (19):(19)$$P\left(x\right)=\int P\left(x,\theta \right)d\theta$$

Therefore, P(x)  can be regarded as the marginal probability of x.

The model m, which is characterized by the model parameter θ, may be represented for a dataset of N data points, where $D=\{x_{1},\;x_{2},\ldots ,\;x_{N}\}$.

The conditional probability of m given D, denoted as P(m│D), can be:(20)$$P\left(m|D\right)=\frac{P\left(D|m\right)*P(m)}{P(D)}$$

This can be calculated for a variety of models m. The optimal model for the given data may be determined by selecting the model with the highest posterior probability. The equation provided can be rewritten in a more academic manner as follows: (21)$$P\left(D|m\right)=\int P\left(D|\theta ,m\right)P\left(\theta |m\right)d\theta$$

The aforementioned equation provides the marginal probability, which is a prerequisite for the computation. It is also possible to make predictions about the likelihood of new, unobserved data points, denoted as $x^{*}$. The conditional probability of $x^{*}$ given D and m, denoted as $P\left(x^{*}|D,m\right)$. The expression may be rewritten in an academic manner as follows in Equations (22) and (23): (22)$$P\left(x^{*}|D,m\right)=\int P\left(x^{*}|\theta \right)P\left(\theta |D,m\right)d\theta$$
(23)$$P\left(\theta |D,m\right)=\frac{P\left(D|\theta ,m\right)*P(\theta |m)}{P(D|m)}$$

The posterior probability of the model parameter θ, given the data D, is calculated using Bayes’ method.

The utilization of Bayesian probability theory enables the representation of varying levels of confidence in uncertain assertions. The derivation of basic probability theory can be achieved by quantitatively representing beliefs, given a limited number of fundamental assumptions [137,138]. The Dutch Book theorem, a significant finding in game theory, asserts that unless our beliefs align with probability theory, including Bayes rule, we will be susceptible to accepting a series of bets known as a Dutch book. These bets are designed in such a way that they are certain to result in financial losses, irrespective of the actual outcomes [135,136].

Bayesian methodologies inherently encompass the principle of Occam’s razor, which asserts that when confronted with two hypotheses that explain all the accessible information, the one that exhibits more simplicity is more probable to be correct. Its practicality extends to other academic disciplines, including but not limited to religion, physics, and medicine. In this analysis, we will examine two distinct models, labeled as $m_{1}$ and $m_{2}$. It is important to acknowledge that $m_{2}$ includes $m_{1}$ as a particular case. For example, linear functions in $m_{1}$ can be seen as specific examples of higher-order polynomials in $m_{2}$. If the data are appropriately represented by model $m_{1}$, such as by the utilization of a linear function, the marginal probability for model $m_{2}$ will be relatively diminished. However, model $m_{2}$ demonstrates greater efficacy in modeling certain datasets, particularly those characterized by nonlinear functions, in contrast to model $m_{1}$. Therefore, it is commonly recognized that Bayesian procedures demonstrate a decreased vulnerability to overfitting concerns, a phenomenon that is typically encountered in other approaches [139].

The effectiveness of Gaussian process-based prognostics relies significantly on the meticulous choice of global and covariance functions. Gaussian process (GP) has historically been employed in the domain of interpolation, where a constant is typically employed to represent a global function. As a result, the global function has been perceived as relatively less significant in comparison to the covariance function. Within the domain of prognostics, which is sometimes used interchangeably with extrapolation, the importance of the global function in GP holds similar weight to that of the covariance function. Nevertheless, the current body of research on the selection or update of the global function to improve the predictive capabilities inside the extrapolation zone is somewhat restricted.

In contrast, covariance functions play a crucial role in determining the quality of GP simulations across large areas. Consequently, several academics have dedicated their efforts to investigating the impact of covariance functions and identifying more effective alternatives. Reference [140] developed three unique optimized neural networks (ONNs) based on the machine learning algorithm (MLA) framework. These ONNs were specifically designed for predicting the FCG rate in various aluminum alloys. The effectiveness of these optimized models was assessed through rigorous testing on different aluminum alloys. Several papers have also proposed the use of nonstationary covariance functions, which may adjust to varying levels of smoothness by including simple covariance functions through addition or multiplication. Reference [141] presented a Bayesian model that addresses the analysis of geographical data with continuous indices. This model has a flexible parametric covariance regression structure, specifically designed for a convolution kernel covariance matrix. 

The determination of the number of input nodes is often found in the data-driven method, since it allows for the inclusion of all relevant information, such as time, loading conditions, and degradation data, as inputs. The study by [142] introduces a novel approach for bearing defect diagnosis through the utilization of a semi-supervised multi-scale convolutional generative adversarial network. This approach leverages both partially labeled samples and an abundant number of unlabeled samples throughout the training process. The findings suggest that the semi-supervised convolutional generative adversarial network, as described, demonstrates favorable performance in the detection of bearing faults. 

## 3. Hybrid Prognostic Approaches

The hybrid prognosis methodology entails the amalgamation of multiple approaches to predict the future state of a certain system. PHM solutions in the aviation sector heavily depend on the exploitation of real-time data to effectively identify potential failures and evaluate the condition of machines. The presented approach is distinguished by its proactive aspect, as it necessitates the application of predictive modeling tools to activate maintenance alerts and anticipate the possibility of faults [2].

Numerous industries have adopted PhM methodologies since they have demonstrated the ability to improve dependability and safety. In the aviation industry, safety requirements are elevated due to the substantial investment and the possible dangers to human life that are linked to aircraft malfunctions or operational disruptions. The utilization of artificial intelligence algorithms is prevalent in commercial operations for the purpose of flight data monitoring systems. Nevertheless, there is a dearth of research that specifically addresses safety-critical systems, such as engine and hydraulic systems [143].

The utilization of hybrid prognosis methodologies within the aviation sector facilitates the augmentation of precision in failure prognostication, thereby making a significant contribution towards the enhancement of dependability and safety within aircraft systems. The incorporation of several methodologies allows for the integration of hybrid prognosis, which in turn facilitates a comprehensive evaluation of the health of a system and enhances the effectiveness of predictive maintenance initiatives [71]. 

The achievement of a hybrid prognosis can be effectively accomplished using several strategies, such as data-driven, physics-based, and knowledge-based techniques. Data-driven techniques utilize previous data to train machine learning algorithms in order to predict future system behavior. Model-based techniques utilize mathematical models to simulate the dynamics of a specific system and offer predictions regarding its future state. Knowledge-based methodologies utilize expert knowledge and pre-established norms to create predictions on the future state of a specific system. The use of a hybrid prognosis has emerged as a feasible strategy in the aviation industry with the aim of effectively monitoring the operational status of various aircraft systems, encompassing engines, hydraulic systems, and avionics. Sensors are utilized to collect real-time data regarding the operating effectiveness of these systems. Following this, the collected data are then subjected to analysis using hybrid prognostic techniques, with the objective of predicting possible problems and scheduling maintenance measures before their actual occurrence [144].

The application of hybrid prognosis in the aviation sector is of considerable importance as a tool for predictive maintenance. The installation of this technology plays a significant role in improving the safety and reliability of aircraft systems while also reducing maintenance costs and operating disruptions. The hybrid prognosis approach deviates from traditional methodologies by incorporating various strategies to predict the future condition of a system’s condition [145]. In contrast, traditional approaches frequently rely on a single strategy for formulating prognostications. An excellent example pertains to the typical prognostic techniques that rely on data-driven approaches, often requiring manual extraction of features from raw sensory input. The process of feature selection can be a laborious undertaking and does not ensure the identification of the most optimal representative attributes in every instance. On the other hand, hybrid prognosis utilizes a hybrid deep neural network structure to extract meaningful features directly from the raw sensory input during the training period. 

By using several approaches, hybrid prognosis possesses the ability to provide a comprehensive understanding of the condition of a system and improve the accuracy of predicting failures. The use of this methodology holds promise for improving the reliability and safety of systems while simultaneously reducing costs related to upkeep and operating disruptions.

One such topic is the development of a hybrid prognostic approach for estimating the RUL of aircraft engines using a combination of PCA, classification and regression trees (CART), and MARS techniques [146]. By employing this fitting method, it is possible to determine the future health condition of a given system and to make precise RUL estimates. The simulation results demonstrate that the PCA-CART-MARS-based methodology could anticipate problems well in advance of their occurrence and accurately predicting the RUL. The primary advantage of the proposed model is its independence from the previous operational conditions of the engine’s input variables. In recent years, multivariate linear regression and ANNs have also been utilized for the RUL prediction. The effectiveness of the PCA-CART-MARS-based methodology was evaluated in comparison to these established methods. The PCA-CART-MARS-based approach has demonstrated tremendous promise in the field of aircraft engine RUL estimation prognostics. The hybrid model employs elements derived from sensor signals to train itself, thereby representing a variety of aircraft engine health states. Lastly, there is a growing interest in prognostics for autonomous electric-propulsion aircraft, which entails predicting and managing potential system failures [147].

The hybrid prognosis technique is an innovative approach that integrates physics-based modeling and data-driven methodologies to improve the accuracy of predictive results. One example of the application of hybrid prognostics in aircraft systems involves the prediction of fatigue life for metallic components located within the structures of aircraft. The authors of the scholarly research paper developed a hybrid prognosis model to accurately predict the crack growth regime and RUL of aluminum components [146]. A supplementary example is available in a study that introduced a novel hybrid prognostic methodology for predicting the RUL of multi-functional spoiler (MFS) systems. The systems are of utmost importance in facilitating the effective operation of aviation spoiler control systems [148]. Another example of the application of hybrid prognosis may be seen in the evaluation of aviation systems, particularly in the analysis of the effectiveness of hybrid electric and distributed propulsion systems integrated into a light aircraft. The primary objective of this study was to assess the importance of electric propulsion systems relative to conventional systems, as well as hybrid propulsion systems. The specific focus was on determining the most appropriate hybrid configuration for light aircraft. The inquiry was carried out using the normalized range factor and range analysis as the main evaluation criterion. Based on the findings of the study, it was concluded that a piston engine is the optimal selection for a hybrid electric propulsion system within the domain of light aircraft. Furthermore, the research findings indicated that a parallel hybrid propulsion system exhibits greater advantages for light aircraft in comparison to a series-hybrid system [149].

There are several techniques available for the merging of physics-based algorithms with data-driven algorithms within the domain of hybrid prognosis. One possible approach is the utilization of a physics-based model to extract features that can then be employed in a data-driven model. One example that may be used to illustrate this concept is the application of a physics-based model to simulate the dynamics of a system under various situations. The resulting data from this simulation can then be used as input for a data-driven model. The data-driven model possesses the capacity to learn information from the given data and afterward generate predictions. An alternate approach entails employing a data-driven model to improve the precision of predictions given by a physics-based model. An example that serves as an illustration involves the application of a physics-based model to produce the first predictions concerning the behavior of a certain system. Following this, the accuracy of these predictions can be further improved by utilizing a data-driven model that has been trained using data collected from the same system. Both approaches employ a physics-based model to include prior knowledge of the system being modeled and enhance the learning process of the data-driven model. The utilization of a combination of physics-based and data-driven approaches has the potential to boost the precision of forecasts when compared to depending simply on either approach independently.

Prognostics technology encompasses a wide range of facets. For instance, prognostics can provide early indications of potential failures and make estimations for the RUL. This can ultimately lead to enhanced availability, reliability, and safety, while also contributing to decreased maintenance expenses. Prognostics, as stipulated in ISO 13381-1 [150], refers to the process of estimating the TTF and associated risk for one or more existing and potential failure modes. This prediction is based on the current condition of the system and its past operational profile [151]. The application of RUL prediction is extensive, encompassing various domains such as military and aerospace systems, manufacturing equipment, constructions, power systems, and electronics [152].

Generally, prediction models for RUL can be classified into the following three categories: experience-based models, data-driven models, and physics-based models, as depicted in Figure 13 [152].

There are various methodologies available for the assessment of system conditions. There are two primary strategies commonly employed for prognosis in aviation, namely, physics-based and data-driven methods. Both techniques include distinct advantages and limits, which is why they are frequently utilized with one another [153]. Prognostics is an academic field that focuses on the prediction of the future performance of a system, with a specific focus on identifying the moment when the system will no longer serve its intended purpose, usually known as its TTF. The RUL factor has a key role in the field of PHM, functioning as a critical element in the decision-making process for maintenance and the mitigation of contingencies. Deterioration is a commonly seen phenomenon that can occur during the whole lifespan of a system or component. Numerous methodologies have been devised to predict the future performance of said systems and determine the threshold at which they will no longer serve their original purpose [152].

### 3.1. Experience-Based Models

Experience-based approaches, also known as knowledge-based approaches, encompass the utilization of historical data collected over a significant period, encompassing failure times, maintenance data, operational data, and other pertinent information, to predict the TTF or RUL. The main advantage lies in their utilization of simple reliability functions, such as the exponential law and Weibull law, instead of complex mathematical models.

The prognostic outcomes provided by these methodologies demonstrate reduced levels of accuracy in comparison to the prognostics offered by physics-based and data-driven approaches, especially in situations when the operational parameters are well-known or when systems are in their initial phases and have limited failure data available [154]. Experience-based models are subject to limitations as they necessitate a substantial amount of personal knowledge pertaining to a certain component or system. In this work, the authors will primarily focus on the contemporary advancements of physics-based and data-driven models in the field of aviation, owing to their widespread usage and appropriateness.

### 3.2. Data-Driven Models

Data-driven methodologies involve the utilization of sensors to acquire online data, which is afterwards transformed into pertinent information. The data are utilized for the purpose of investigating degradation patterns using a range of models and tools, including NNs, Bayesian networks (BNs), and Markovian processes, as well as statistical methods. This analysis aims to forecast the future health condition and the accompanying RUL of the system.

Data-driven approaches possess a notable advantage over both physics-based and experience-based methods. This advantage stems from the fact that, in practical industrial applications, obtaining trustworthy data is a more feasible task compared to developing models that capture physical or analytical behaviors. Furthermore, the behavioral models derived from actual monitoring data yield more accurate predictive outcomes compared to those derived from historical data as mentioned in the previous chapter.

### 3.3. Physics-Based Models

The physics-based methodologies employ an analytical framework that incorporates a set of differential or algebraic equations to effectively represent the dynamic characteristics and deterioration of the system. Ref. [155] presented a pioneering study where authors introduced a fatigue life model for ball bearings that relies on stress analysis. The model has improved accuracy in predicting prognosis. Nevertheless, it is crucial to acknowledge that real-world systems often exhibit nonlinearity, and the degradation mechanisms associated with them tend to be inherently stochastic. As a result, the application of analytical models presents difficulties. Hence, the feasibility of using this approach may be limited as aforementioned in Section 2.1. 

### 3.4. Hybrid Prognostic Models

Hybrid prognostic methodologies integrate the advantageous aspects of both data-driven and physics-based modeling to enhance the precision and reliability of failure prognostications. Various methodologies can be employed to anticipate and mitigate failures in advance. This can be achieved by carefully picking pertinent hIs and calculating the PDFs of hIs under both optimal and deteriorated conditions [34,156,157]. Hybrid prognostic methodologies may encompass the integration of physics-informed machine learning techniques [158].

One advantage of hybrid prognostics is its capacity to enhance the precision and dependability of failure forecasts through the integration of the robustness of data-driven and physics-based modeling [159]. The hybrid strategy, which combines elements of physics-based and data-driven methodologies, is employed to mitigate the limitations of each approach and leverage their respective advantages. However, it is important to note that the hybrid approach does retain certain downsides associated with both methods. 

The flowchart proposed by [12] serves as a reference for the purpose of selecting an appropriate hybrid prognostics strategy, as depicted in Figure 14. 

The section presents a range of ways that utilize different combinations of the three categories. The picture presented in this context is derived from a study conducted by [160] and visually represents five unique combinations.

**Hypothesis 1.** *Posits the integration of an experience-based model and a data-driven model*.

**Hypothesis 2.** *Posits the integration of an experience-based model and a physics-based model*.

**Hypothesis 3.** *posits that the utilization of a data-driven model in conjunction with another data-driven model can yield significant benefits*. 

**Hypothesis 4.** *Posits the integration of a data-driven model and a physics-based model*.

**Hypothesis 5.** *postulates that the amalgamation of an experiential model, a data-centric model, and a physics-oriented model will yield favorable outcomes*.

One instance of a prognostic application within the aircraft sector can be observed in the Joint Strike Fighter (JSF) [161]. The system is intended for utilization by the United States Air Force, Navy, and Marine Corps, as well as select allied nations inside the United States’ sphere of influence. The existing strategy entails implementing a PHM system that offers fault detection and isolation capabilities for all significant systems and subsystems present on the aircraft. Additionally, the system will proceed with prognostics specifically for certain components. PHM constitutes a pivotal factor in substantiating the selection of a singular-engine aircraft, with the primary objective of enhancing safety measures and diminishing maintenance expenses. The architectural design being proposed incorporates an off-board PHM system, which will utilize data mining methodologies [162,163]. Moreover, automated prognostic research has been implemented in a diverse range of systems, encompassing actuators, aerospace structures, aircraft engines, batteries, bearings, clutch systems, cracks in rotating machinery, electronics, gas turbines, hydraulic pumps and motors, military aircraft turbofan oil systems, semiconductor manufacturing, heating, ventilation, and air conditioning, wheeled mobile robots, and UAV propulsion.

Prognostics applications have the capability to operate either in real-time, or near real-time, regardless of whether they are aboard or off-board. Prognostics can also be implemented in an offline manner, independent of the operational duration of the system being monitored. Real-time prognostics involve the utilization of online data obtained from a data-gathering system to estimate the RUL of a system. This calculation enables the system to provide a timely warning regarding an imminent failure, hence facilitating the reconfiguration of the system and the planning of subsequent missions. The offline prognostics system utilizes fleet-wide system data and conducts extensive data mining procedures that are not feasible to be executed onboard in real time due to resource limitations and time constraints. The utilization of outcomes derived from an offline prognostics system holds the potential for informing maintenance planning and facilitating decision-making processes within the realm of logistical support management. The application of prognostics first revolved around the practice of forecasting [164].

#### 3.4.1. Physics-Based Models or Data-Driven Models

The data-driven model involves gathering monitoring data from sensors to simulate the system’s degradation. The data are subjected to pre-processing to find relevant elements that can be utilized in the development of models for health assessment and prediction of RUL. Several machine learning techniques can be identified as aforementioned, including NNs, HMMs, regression analysis, and support vector regression (SVR).

Physics-based model necessitates a comprehensive comprehension of the underlying physical system, encompassing the intricate dynamics of degradation through time. The utilization of physical principles is employed in the construction of a system model that is afterward utilized for the purpose of simulations and RUL prediction [165]. The study conducted by [166] employed a mathematical approach to investigate the deterioration of a vehicle’s suspension system, with a focus on physics-based prognostics. In the investigation of the progression of damage in a two-well magneto-mechanical oscillator, the authors [167] adopted a comparable methodology by suggesting a technique rooted in the principles of dynamic systems. It is crucial to recognize that within the realm of physics-based prognostics, the construction of a model necessitates the inclusion of a degradation model, which encompasses factors such as fatigue, corrosion, or wear-induced cracks. It can be observed that data-driven models produce results with lower levels of precision than physics-based methods. Accessibility of the data used to train deterioration models is an additional limitation associated with data-driven prognostics. As stated previously, it is necessary to collect data that accurately represents the degradation’s behavior. In practical applications, it is essential to keep in mind that the statistics pertinent to the deterioration of assets under identical operating conditions may exhibit variation. The model derived from this dataset will capture the mean value; consequently, estimates of RUL will be imprecise [168].

Practitioners tend to prefer data-driven solutions due to their cost-effectiveness, versatility, and simplicity. The utilization of physics-based methodologies in industrial systems is impeded by the inherent difficulty of building a physical model that precisely depicts the deterioration of the system. Physics-based approaches can be utilized in systems that have pre-existing models or in certain types of systems, such as mechatronic systems. However, it is necessary to perform empirical investigations to determine the underlying patterns of system degradation. One notable benefit of employing physics-based methodologies is their capacity to yield precise forecasts, especially in scenarios characterized by restricted data availability. Nonetheless, the process of constructing precise models can present difficulties and necessitates a profound comprehension of the system being subjected to modeling. In contrast, data-driven methodologies exhibit a higher level of ease in their implementation, albeit necessitating a substantial volume of historical data for optimal efficacy. Table 7 presents a comprehensive analysis of the advantages and disadvantages associated with each approach.

In essence, physics-based methodologies employ mathematical models grounded in fundamental physical principles to provide predictions, whereas data-driven methodologies depend on historical data to acquire knowledge of the system’s behavior and generate forecasts. Both methodologies possess their respective merits and demerits, and the selection between them is contingent upon the application and the accessibility of data. The integration of both approaches in a hybrid approach to prognostics can effectively harness their inherent strengths, resulting in improved forecasting capabilities. Concurrently, the hybrid approach possesses the capacity to alleviate the specific drawbacks encountered by individuals [169]. Ensemble learning holds promise for the future use of amalgamating and incorporating diverse data-driven prognostics techniques. The integration of online algorithms and uncertainty poses a significant concern in the context of a hybrid strategy for RUL estimation [168,169].

Hybrid prognostics that use a combination of physics-based and data-driven techniques can be classified into four discrete categories [170], as outlined in Table 8.

#### 3.4.2. Hybrid Approach Integrating Data-Driven Models and Physics-Based Models

There does not exist a flawless prognostics model. Every model has its own set of pros and limitations, and it is more prudent to consider the appropriateness of a model based on the specific scenario being analyzed. In the field of power devices, a research study conducted by [171] introduced a methodology for predicting RUL by employing a combination of data-driven and physics-based prognostics algorithms. The drain-source ON-state resistance was employed by the researchers as a metric for assessing the health status. The methodology employed in this study involved the utilization of Gaussian process regression as the data-driven component. The methodologies employed in this study encompassed two physics-based techniques, namely, an EKF and a PF. The researchers conducted accelerated aging experiments on power devices and employed prognostic performance indicators to evaluate and compare the outcomes of the different methodologies. The PF approach exhibited superior performance in the field of prognostics. The superiority of the physics-based approach can be attributed to the utilization of an exponential deterioration model with two parameters that are computed online within a Bayesian framework. The Gaussian process regression model, which relies on data-driven techniques, was unable to generate accurate RUL forecasts until a distinct degradation behavior, namely, of exponential nature, became evident. On the other hand, in cases when the degradation does not conform to an exponential model due to factors such as noisy data or varying failure modes, the utilization of a data-driven model would provide more precise outcomes. This is because the findings obtained from the data-driven model may be compared to the historical deterioration patterns [160].

In the realm of RUL analysis, it is widely observed that a considerable number of researchers exhibit a preference for a hybrid methodology. This approach entails the integration of both data-driven and physics-based models, with the intention of capitalizing on the unique advantages offered by each model type. The objective behind this integration is to enhance the accuracy and reliability of RUL prediction [160].

#### 3.4.3. The Prognostics Fusion Framework

The incorporation of data-driven and physics-based approaches into a fusion prognostic framework shows potential for accurately predicting RUL. The physics-based technique, also known as the model-based approach [160], entails leveraging an understanding of the fundamental principles of physics to generate accurate estimations. The approach outlines the procedure for the degradation of a system using an analytical equation known as a degradation model. The degradation model should accurately represent the process of degradation; nevertheless, in practical implementation, deviations from the model may occur. The use of data-driven prediction methods, which utilize past data and the specific system being studied, holds promise for improving the accuracy of forecasts and reducing uncertainty. The fusion architecture of merging data-driven and physics-based models is illustrated in Figure 15. 

The figure depicted above provides a preliminary representation of the intricate relationship that exists between data-driven and physics-based methodologies, as referenced in [152]. The integration of prognostics incorporates two data-driven techniques into the traditional physics-based PF architecture, hence enhancing the precision of predictions. The paper presents a novel approach utilizing data-driven techniques for estimating the measurement model and another for forecasting future measurements in long-term prediction scenarios.

The hybrid prognostics framework extends the applicability of Bayesian state estimation by incorporating two data-driven techniques into a physics-based approach, namely, utilizing the PF method. The sensor readings $Y_{k}$ typically do not provide direct access to the internal system state $X_{k}$, such as degradation, in a complex system. This necessitates the utilization of a physics-based approach to indirectly estimate the internal state of the system. The conventional Bayesian state estimation method is based on an analytical measurement model, $Y_{k}=h\left(X_{k}\right)+v_{k}$. 

However, in many instances, it is not possible to obtain an analytical representation of the measurement model. As a result, an approach based on data analysis is employed instead. The utilization of the estimated data-driven measurement model enables the execution of state tracking in a conventional manner, employing the system degradation model $Y_{k}=h\left(X_{k-1}\right)+w_{k}$. During the phase of state prediction, the PF based on classical physics utilized the system degradation model to extrapolate the internal state of the system. In the proposed framework for fusion prognostics, a secondary data-driven approach is employed to forecast future measurements as in Equation (24):(24)$$\hat{Y_{k+1}}=g(X_{k},Y_{k-1},\ldots )+u_{k+1}$$

The inputs are reintroduced into the PF algorithm. The state prediction phase is conducted similarly to the state tracking phase, utilizing the anticipated measurements. The particles and their corresponding weights can then be further adjusted [152].

#### 3.4.4. Limitations of Hybrid Prognostic Approaches

Nevertheless, like any other methodology, hybrid prognostics have inherent limitations. One limitation is the challenge associated with developing hybrid models that effectively combine the benefits of data-driven and physics-based modeling techniques [34,172]. The effective integration of the distinct benefits provided by data-driven and physics-based modeling methodologies represents an additional obstacle in the development of hybrid prognostic models in the aviation domain. 

Hybrid models are required to possess the ability to accurately predict and prevent failures in advance. This is achieved by carefully selecting the most relevant hIs and accurately calculating the PDFs of hIs for both normal and deteriorated states [157,173]. Another challenge arises from the intricacy of hybrid models, which requires a significant level of expertise for their creation and continuous upkeep. Additionally, it is important to acknowledge that hybrid models may require a significant amount of data for training and validation, which can be difficult to do in certain situations [174].

To tackle these challenges, researchers are presently involved in the exploration of innovative methodologies that combine data-driven and physics-based modeling techniques. Furthermore, their efforts are focused on improving the accuracy and reliability. Within the domain of hybrid prognostics for aircraft systems, PDFs can be utilized to evaluate the probability of a system or component existing in a particular state, such as being in a state of optimal health or experiencing degradation. This estimation is derived by considering the hIs linked to the system or component. Hybrid models possess the capacity to accurately predict and prevent failures by employing the estimate of PDFs for hIs in both healthy and degraded states.

For instance, consider a system comprising two hIs, namely, temperature and vibration. The PDFs of the hIs for both healthy and impaired states can be approximated using historical data. Subsequently, upon the acquisition of novel temperature and vibration data pertaining to the system, the PDFs can be employed to approximate the likelihood of the system being in either a sound or deteriorated condition. If the likelihood of the system being in a deteriorated state is considerable, proactive measures can be implemented to mitigate the risk of a failure prior to its manifestation.

Current studies in hybrid prognostics for aviation have been primarily dedicated to the advancement of novel approaches that integrate data-driven and physics-based modeling techniques. The objective is to enhance the precision and dependability of failure prognostications. An investigation was conducted in a study to examine the utilization of classification methods, as opposed to regression approaches, for the purpose of identifying problems in aircraft systems [34]. Recent papers examined the current advancements in hybrid electric aircraft and the techniques employed for managing their energy [158,175].

These methodologies possess the capability to enhance the safety and reliability of aircraft systems through the precise anticipation and preemptive mitigation of faults prior to their occurrence. A study was conducted to examine the integration of physics-based and deep learning models for the purpose of prognostics [176]. The objective of this study is to construct hybrid models that proficiently integrate the advantages of data-driven and physics-based modeling to precisely forecast and preempt problems prior to their occurrence [12].

#### 3.4.5. State of the Art

Reference [2] focuses on prognostics in aircraft systems and presents an overview of contemporary research pertaining to predictive maintenance (PM) techniques employed for the hydraulic system and engine of airplanes. The authors examine the significance of PM and cutting-edge data pre-processing techniques in the context of handling huge datasets. Additionally, they ascertain emerging patterns and obstacles within the realm of project management for aircraft systems. This work presents a thorough examination of the existing body of research in this field and serves as a great reference for individuals seeking to expand their knowledge about PM for aircraft systems [2,177].

Reference [4] examines the latest advancements in research and the practical uses of prognostics modeling methods in the field of engineering systems. The study that has been examined is categorized into three primary domains, depending on whether they integrate the understanding of the physics of failure into prognostics. These domains are the data-driven prognostic techniques, the physics-based prognostic methods, and the hybrid prognostic approaches. The technical advantages and limitations of each prognostic approach are analyzed and explained. Additionally, this paper provides a comprehensive overview of the research and technological obstacles encountered in the field of engineering system prognostics. Furthermore, it identifies potential directions for future research endeavors.

Reference [178] elucidates the creation of a novel and authentic dataset comprising run-to-failure trajectories for a fleet of aircraft engines operating in actual flight conditions. This dataset holds significant value for the field of prognostics and diagnostics. The dataset utilized in this study was derived from the Commercial Modular Aero-Propulsion System Simulation (CMAPSS) model, which was originally designed by NASA. The authors emphasize the significance of possessing representative run-to-failure datasets to facilitate the creation of data-driven prognostics models. They also highlight the versatility of their dataset, which may be utilized for both prognostics and fault diagnostics purposes. The paper offers significant insights into the creation of authentic datasets for prognostics in aircraft systems, rendering it a helpful resource for individuals seeking relevant information in this field. 

The utilization of data-driven methodologies, particularly ML, has significantly advanced maintenance modeling in recent times, leading to a wide array of practical applications [179]. In this study, several conclusions can be drawn. Firstly, the utilization of publicly accessible data has the potential to enhance research endeavors. Secondly, a significant proportion of academic papers depend on supervised techniques that necessitate annotated data. Thirdly, the amalgamation of multiple data sources has the potential to enhance the accuracy of results. Lastly, the adoption of deep learning methods is expected to rise, but it is contingent upon the development of efficient and interpretable approaches as well as the availability of substantial quantities of labeled data. ML methodologies are currently being utilized for the purpose of mechanical defect detection and prediction within the framework of practical industrial manufacturing scenarios [180]. This analysis demonstrates that there has been a growing number of studies conducted in the manufacturing business in recent years. However, further research is required to effectively tackle the issues posed by real-world situations [180,181].

Reference [182] examines the academic and industry literature to identify the primary technological domains of electric aviation. These domains include battery technology, electric machine technology, airframe technology, and propulsion technologies. The paper discusses the current state of these technologies, their projected advancements in the future, as well as the challenges they face. This study examines several design concepts, prototypes, and current electric aircraft products to identify the limitations posed by technology progress and regulatory frameworks that may hinder the implementation and commercialization of suggested electric airplanes.

Reference [175] provides an overview of the current research progress in the field of hybrid aircraft design and energy management, as well as hybrid propulsion systems. Another instance of a hybrid methodology for prognostics can be observed in the context of micro-electromechanical systems (MEMS), as elucidated by [183]. The proposed methodology consists of the following two distinct stages: an initial offline phase dedicated to the characterization and modeling of the MEMS degradation, and a subsequent online phase where the derived degradation model is employed in conjunction with the available data for prognostic purposes. The offline phase encompasses the utilization of physics-based models to elucidate the behavior of the MEMS and its constituent parts. Conversely, the online phase entails the application of data-driven techniques to revise the model parameters and generate forecasts regarding the system’s RUL.

An additional illustration may be found in the form of a model-based hybrid technique utilized for circuit breaker prognostics. This approach effectively integrates the continuous and discrete temporal behavior of the system, as demonstrated in references [184,185]. This combination is well-suited for applications that need the consideration of deterministic system behavior, particularly in cases where the deterioration is observed to increase at specific discrete time intervals. The instances serve as mere illustrations of the potential applications of hybrid methodologies in the field of prognostics. The integration of data-driven and physics-based methodologies provides a robust approach to effectively forecast the future dynamics of a given system, capitalizing on the respective advantages of both approaches.

Reference [186] examines the application of Integrated System Health Management (ISHM) technology in aerospace systems. ISHM technology integrates sensor data and historical state-of-health information of components and subsystems to deliver actionable insights and facilitate intelligent decision-making pertaining to system operation and maintenance. The core foundation of ISHM is predicated on the utilization of evaluations and prognostications pertaining to the overall well-being of a system. This encompasses the timely identification of malfunctions and the calculation of the remaining duration of optimal functionality. Various reasoning techniques, such as model-based, data-driven, or hybrid approaches, can be employed to optimize the promptness and dependability of diagnostic and prognostic data. 

Reference [187] presents a novel approach in the field of artificial intelligence, utilizing two distinct neural networks known as the growing neural networks (GNN) and variable sequence LSTM (VarLSTM) model. The objective of this study is to automate the complex tasks of diagnosis, prognosis, and health monitoring (DPHM) specifically for aerospace systems. The proposed model utilizes the residuals between the measured telemetry data and the predictions generated by the GNN algorithm to estimate a HI value. Subsequently, this HI value is extrapolated for prognostics.

Reference [188] introduces a PHM model that integrates various deep learning methods to perform condition assessment, fault classification, sensor prediction, and RUL estimation for aviation systems. In this study, a recurrent network utilizing LSTM is employed to forecast numerous multivariate time series originating from sensors. Additionally, a deep belief network (DBN) is utilized to evaluate the condition of the system and classify errors pertaining to aviation systems. The estimation of the RUL can be achieved by integrating condition assessment and sensor prediction techniques.

Another instance of a hybrid prognostics technique was suggested and implemented in a study involving battery degradation, with the aim of demonstrating the possible advantages associated with the hybrid prognostics approach [189]. An additional illustration pertains to a hybrid prognostics methodology employed to assess the RUL of wind turbine bearings [173]. A novel approach to PHM has been suggested, which integrates data-driven and physics-of-failure models. This methodology aims to address fault diagnosis and life prediction of electronic equipment [190]. Table 9 presents some recent outstanding advancements in research pertaining to hybrid prognosis within the aviation sector.

## 4. Discussion

Overall, it appears that there is a considerable body of significant research being undertaken in this field. Researchers are now engaged in the exploration of innovative approaches that involve the application of machine learning, physical principles, and other data-driven techniques to improve the effectiveness of prognostic tactics for PM and hybrid systems in PHM for critical aircraft systems. These investigations have been conducted over many instances. Hybrid prognostic approaches have gained widespread use across several industries, encompassing sectors such as battery manufacturing, wind turbine bearing production, and electronic device development. 

However, these principles are only applicable to a limited number of systems in real-life scenarios, primarily because they have undergone a relatively limited development period compared to other more extensively established engineering systems. Furthermore, it is important to note that there is currently no universally applicable prognostic model that can be employed across all systems. It is crucial to recognize that the selection of an appropriate prognostic model should be contingent upon the specific characteristics and requirements of the individual systems under consideration. A further concern that warrants attention is the absence of a readily available dataset for researchers to utilize in evaluating the efficacy of their models. There remains significant scope for further advancements in the domain of PHM for aviation systems, with particular emphasis on hybrid prognostic models.

## 5. Conclusions

The proper implementation of PHM plays a vital role in ensuring the safety and reliability of aircraft. PHM systems employ diverse datasets for the purpose of diagnosing potential failures and predicting the health of machinery and facilitate the proactive identification and mitigation of potential failures. The application of PHM in aircraft systems has yielded positive outcomes; however, there exists a notable research deficiency in the integration of hybrid PHM applications. Notwithstanding these challenges, PHM presents a multitude of advantages and benefits. However, it is imperative to address the obstacles to attain optimal performance. This paper presented a comprehensive examination of the existing research progress on PHM in the aviation sector. It highlighted a range of widely used algorithms and their respective applications, while also discussing the benefits and drawbacks associated with each algorithm. The paper included a limited selection of methods and algorithms, as the authors had to take into account the constraints of paper length. The authors of this paper will persist in their efforts to advance the development of physics-based and data-driven models to achieve the ultimate objective of integrating a groundbreaking hybrid prognostic approach for aircraft systems. 

## Figures and Tables

**Figure 2 sensors-23-08124-f002:**
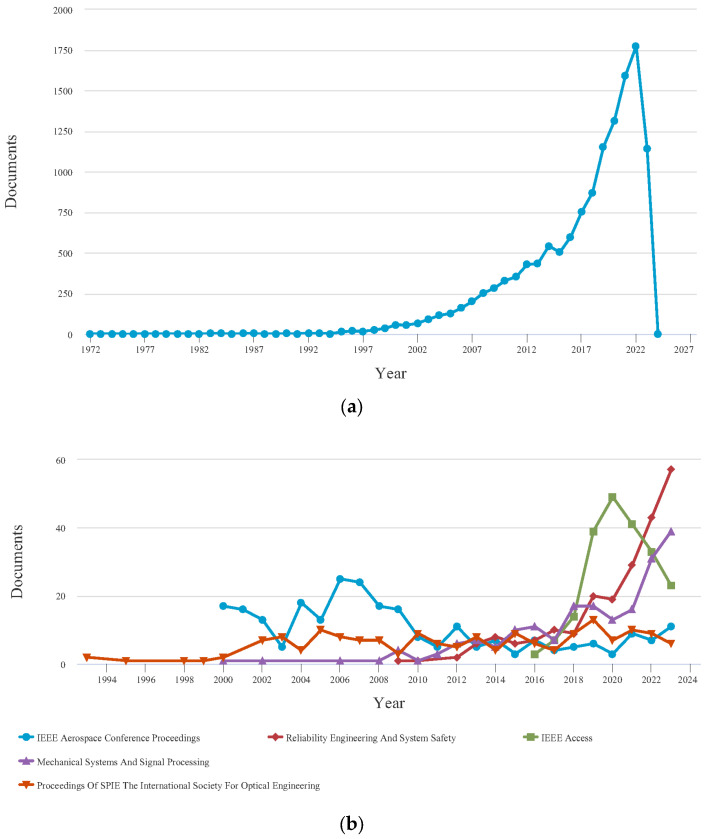
Publication results with keywords ‘prognostic’ ‘aircraft’ ‘system’. (**a**) Documents by year. (**b**) Documents by sources.

**Figure 3 sensors-23-08124-f003:**
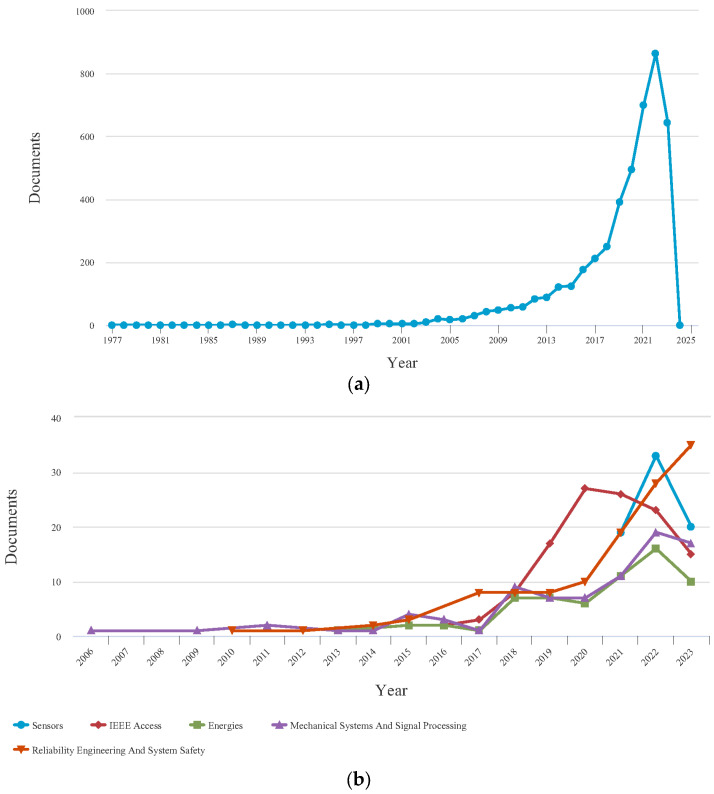
Publication results with keywords ‘hybrid prognostic’ ‘aircraft’ ‘system’. (**a**) Documents by year. (**b**) Documents by sources.

**Figure 4 sensors-23-08124-f004:**
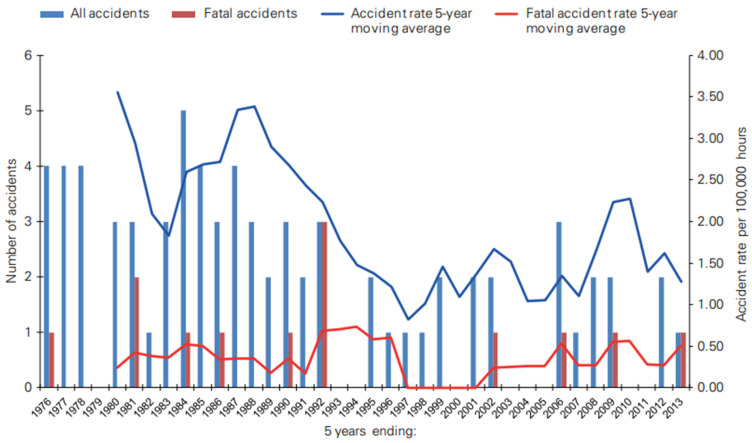
Chronology of reportable accidents (rate per 100,000 flight hours) (from CAA, CAP 1145 Offshore helicopter review).

**Figure 5 sensors-23-08124-f005:**
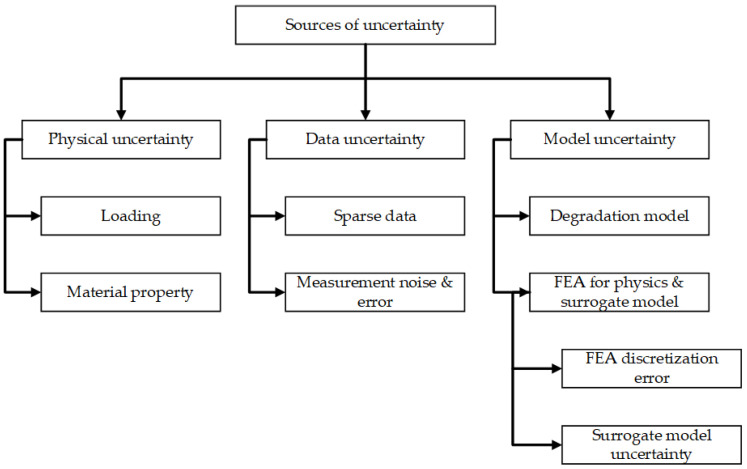
Classification of uncertainty in the prognostics.

**Figure 6 sensors-23-08124-f006:**
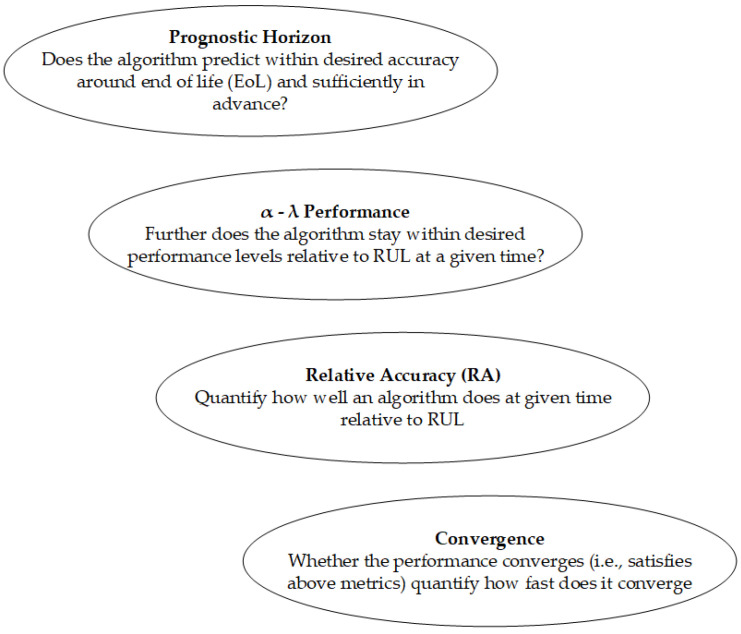
Steps for determining prognostic metrics (Adopted from [20]).

**Figure 7 sensors-23-08124-f007:**
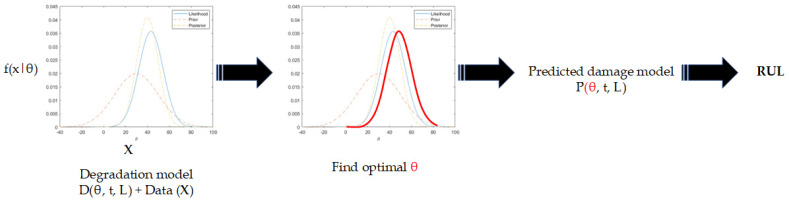
Illustration of physics–based model.

**Figure 8 sensors-23-08124-f008:**
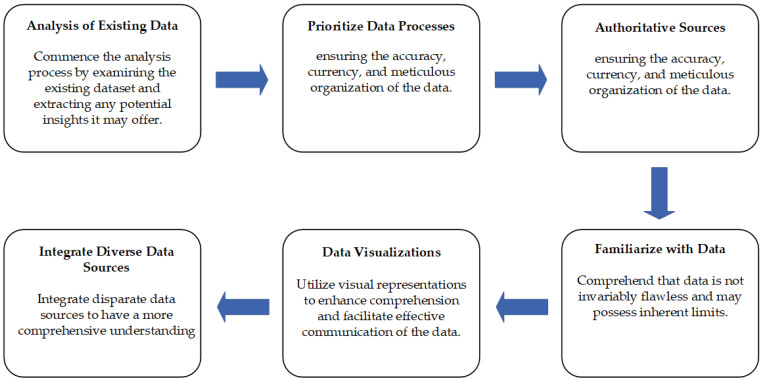
Execution steps of a data-driven approaches.

**Figure 9 sensors-23-08124-f009:**
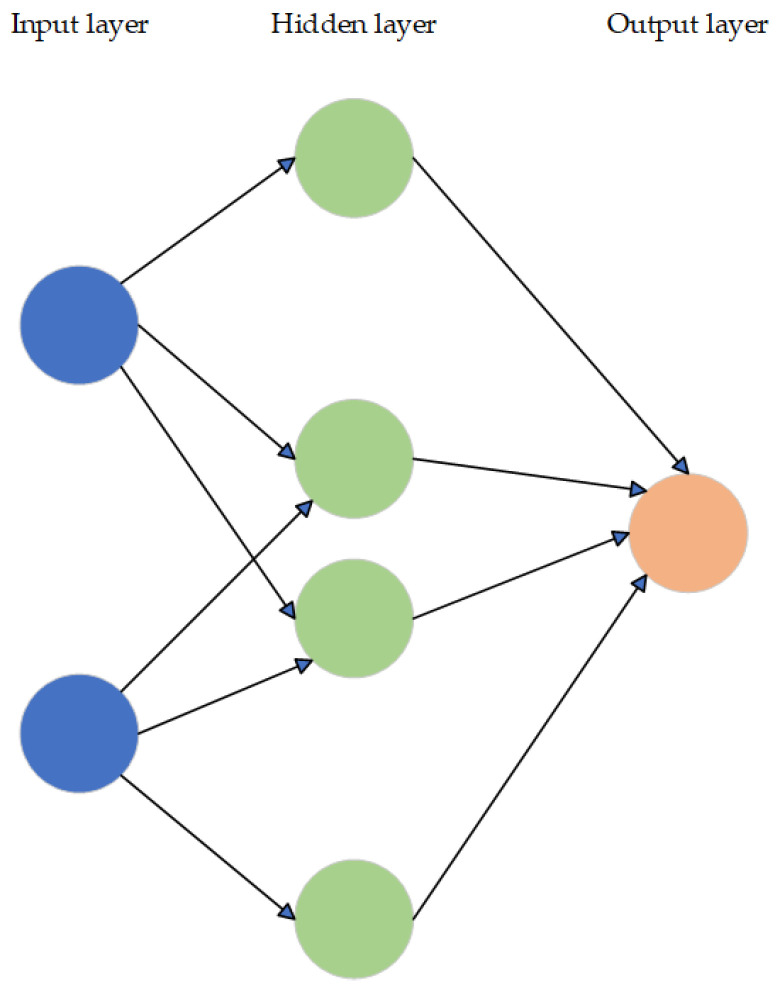
Simplified view of a feedforward ANN.

**Figure 10 sensors-23-08124-f010:**
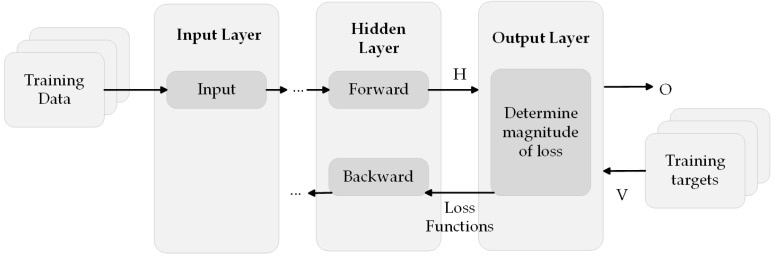
Illustration of the data flow within a neural network, including the subsequent output layer.

**Figure 11 sensors-23-08124-f011:**
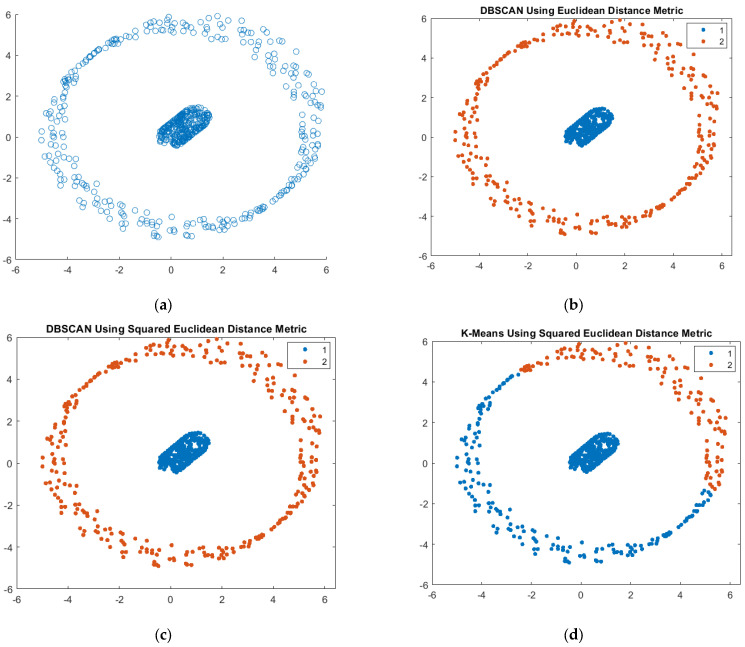
Illustration of how DBSCAN and k-means clustering work. (**a**) Dataset contains two distinct clusters. (**b**) DBSCAN correctly identifies the two clusters with Euclidean distance. (**c**) DBSCAN correctly identifies the two clusters with squared Euclidean distance. (**d**) K-means clustering fails to correctly identify the two clusters using squared Euclidean distance.

**Figure 12 sensors-23-08124-f012:**
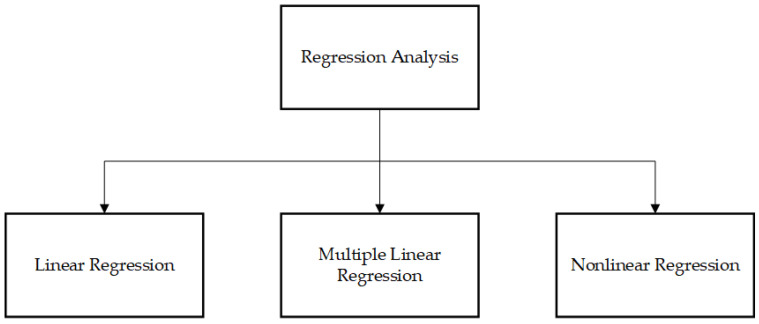
Regression analysis classification.

**Figure 13 sensors-23-08124-f013:**
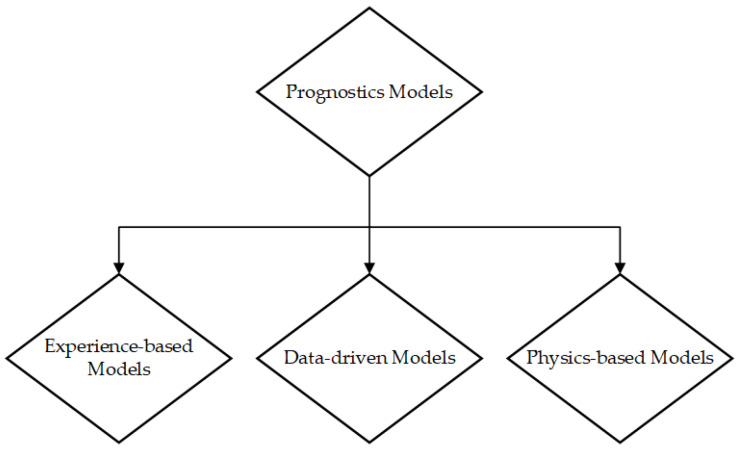
Classification of prognostic models.

**Figure 14 sensors-23-08124-f014:**
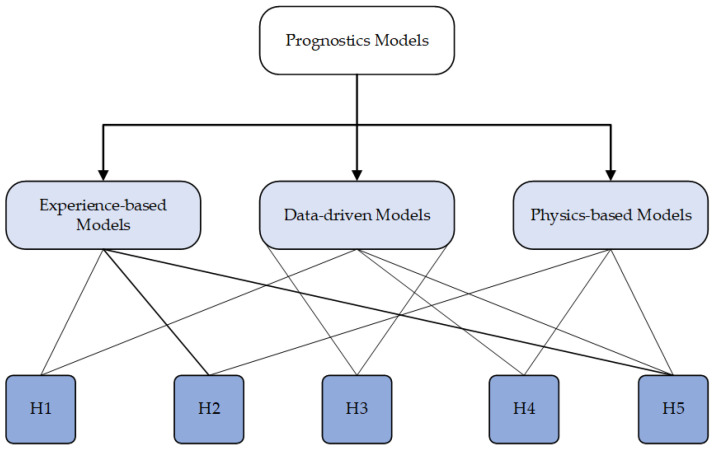
Hybrid prognostic models (adopt from [160]).

**Figure 15 sensors-23-08124-f015:**
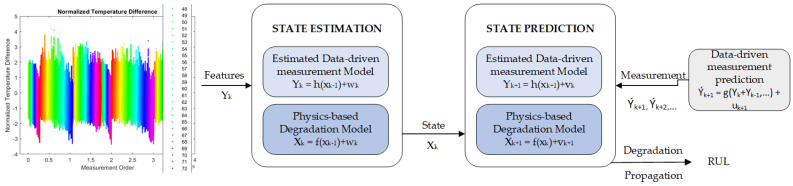
Data-driven and physics-based methods fusion prognostics framework.

**Table 1 sensors-23-08124-t001:** Comparative analysis of CBM and conventional maintenance methods.

	CBM	Conventional Maintenances
Trigger Mechanism	Maintenance strategy initiated either by real-time monitoring or periodic evaluation of an asset’s state. This can encompass the utilization of diverse sensors and monitoring methodologies to gather data pertaining to variables such as temperature, vibration, oil quality, and other relevant parameters.	Involves scheduling maintenance work based on predefined time intervals or consumption levels. RM responds to failures after their occurrence.
Temporal Aspect	Implemented exclusively when there is observable evidence indicating the degradation or imminent critical state of the asset’s condition. By minimizing needless maintenance, the longevity of the asset is extended.	Rise to the issue of over-maintenance when operations are carried out preemptively before required. Relying only on RM can lead to expensive periods of downtime and repairs due to unforeseen problems.
Cost Efficiency	Typically considered to be a more economically efficient approach due to its focused maintenance operations that are informed by the actual state of assets. It effectively reduces downtime and minimizes the occurrence of superfluous maintenance tasks.	Might incur significant expenses because of unneeded maintenance efforts. Conversely, relying on RM can lead to increased repair costs and productivity losses.
Operational Lifespan	Aimed at optimizing the lifespan of assets by implementing maintenance activities at the most opportune moments to effectively prolong the asset’s operational duration.	Potentially result in the degradation of assets because of excessive maintenance (PM) or premature failure owing to inadequate maintenance (RM).

**Table 2 sensors-23-08124-t002:** The comparison between unsupervised learning and supervised learning.

Properties	Unsupervised Learning	Supervised Learning
Definition	Unsupervised learning is when a machine learns without being watched by a person. A machine searches through data on its own for trends.	Supervised learning is a type of machine learning that happens with the help of a person. Input data are labeled with answer keys that show the machine how to obtain the results that are wanted.
Data	Unlabeled	Labeled
Utilization of data	A model only has input factors and no output data to go with them.	A model is given input variables, output variables, and an algorithm to learn the function from inputs to outputs.
When to apply	The user lacks a clear understanding of the specific criteria they are seeking inside the dataset.	The user possesses a clear understanding of the desired attributes inside a dataset.
Useful for	Clustering and association problems	Classification and regression problems
Accuracy	Deliver less precise outcomes	Provision of more precise outcomes
Algorithms	K-means Gaussian mixture models Frequent Pattern (FP) growth Principal Component Analysis (PCA)	Support Vector Machines (SVM) Decision trees Random Forest (RF) Naïve Bayes
Use cases	Recommender systems Anomaly detection	Image recognition Demand forecasting

**Table 3 sensors-23-08124-t003:** Advantages and limitations of SVM.

Advantages	Limitations
The method demonstrates efficiency in high-dimensional environments, even in scenarios where the number of dimensions surpasses the number of samples.	There is a chance of overfitting when the number of features is much higher than the number of samples. When this happens, it is important to choose the right kernel functions and regularization terms.
The decision function has the potential to be tailored to specific needs by the utilization of alternative kernel functions, hence enhancing its versatility.	SVMs do not inherently provide estimation and probability values. These values must be computed using five-fold cross-validation, a computationally intensive method.
The decision function has the potential to be tailored to specific needs by the utilization of alternative kernel functions, hence enhancing its versatility.	

**Table 4 sensors-23-08124-t004:** Machine learning techniques and its working scheme.

Methods	Working Scheme
Artificial Neural Networks (ANNs)	ANNs are composed of interconnected nodes or neurons that process information using a connectionist paradigm. The complex structure and cognitive processes of the human brain serve as their foundation. By modifying how neuronal connections are made, ANNs are able to learn new information and identify patterns in large datasets.
Support Vector Machines (SVMs)	SVMs determine how to partition the data into distinct classes by identifying the hyperplane that does so most effectively. SVMs employ a technique known as the ‘kernel trick’ to transfer the data into a higher-dimensional space where it is simpler to locate a distinct hyperplane.
Decision Trees (DTs)	Decision trees function by continually dividing data into groups based on the values of the features that comprise the tree. At each split, the optimal method to divide the data into distinct classes is determined based on a particular characteristic. Each leaf node of the resulting tree structure represents a class name.
K-means	K-means clustering is an unsupervised learning technique that divides data into k categories, where k is a user-specified parameter. Each data point is assigned to the cluster with the closest mean, and the cluster means are updated until convergence.

**Table 5 sensors-23-08124-t005:** Machine learning techniques and its advantages and disadvantages.

Methods	Advantages	Limitations
Artificial neural networks (ANNs)	Useful tools for modeling complex input—output relationships. Accurate predictions even when the underlying relationships are nonlinear.	Hard to interpret. Require massive data. Computational cost
Support vector machines (SVMs)	Effective at locating the optimal class boundary. Robust over overfitting	Dependent on the selection of kernel function and additional hyperparameters May not scale well to very large datasets
Decision trees (DTs)	Easy to interpret both numerical and categorical data Relatively quicker to train even with smaller datasets	Prone to overfitting specifically when the tree is set to grow too deep
K-means	Simple and efficient for partitioning data into clusters Capable of handling unlabeled data and large datasets	User must specify the number of clusters May not function well with clusters of varying sizes

**Table 6 sensors-23-08124-t006:** The components of a hidden Markov model (HMM) and its corresponding states.

**Components**	**States**
$X_{n}$	is a Markov process whose behavior is not directly observable
$P\left(Y_{n}\in A|X_{1}=x_{1},\ldots ,\;x_{n}\right)=P\left(Y_{n}\in A|X_{n}=x_{n}\right)$	For every $n\;\geq 1,\;x_{1},\ldots ,\;x_{n}$
**Components**	**States**
$X_{t}$	is a Markov process whose behavior is not directly observable
$P\left(Y_{t0}\in A|{\left\{X_{t}\in B_{t}\right\}}_{t\leq t_{0}}\right)=P\left(Y_{t0}\in A|X_{t0}=B_{t0}\right)$	For every $t_{0}$, and every family of sets ${\left\{B_{t}\right\}}_{t\leq t_{0}}$

**Table 7 sensors-23-08124-t007:** The comparison between data-driven approaches and physics-based approaches in various academic disciplines.

Approaches	Advantages	Limitations
Physics-based Models	Enhanced accuracy Deterministic methodologies System-centric approach The dynamics of the states can be calculated and forecasted at every instance The failure thresholds can be established based on the performance of the system Ability to simulate several deteriorations	The utilization of a degradation model is necessary The implementation cost is notably high. The application of the degradation model on complex systems poses challenges.
Data-driven Models	Simple implementation Low cost	Reliable sources of dataset Results vary even under same operating condition Less accuracy Challenging to consider the effects of varying operating conditions. The focus is mostly on individual components rather than the system as a whole. Establishing failure thresholds is a complex task.

**Table 8 sensors-23-08124-t008:** Overview of four distinct types of fusion between physics-based and data-driven models.

Types	Objectives	Description
Type 1	Utilizing the data-driven approach for inferring the physical model	In situations where obtaining the degradation model is challenging or direct measurement of the system state is not feasible, a data-driven model can serve as a viable alternative to a complex physics-based model.
Type 2	Utilizing a data-driven approach for the purpose of estimating forthcoming measurements in conjunction with the physics-based method	When there is a scarcity of measurements for long-term prediction, the data-driven technique can provide predicted measurements that can be considered as additional measurements inside the physics-based method
Type 3	Applying the data-driven approach to estimate and alter the parameters of the physics-based methodology	The data-driven approach refers to a methodology that utilizes data to analyze the correlations and patterns associated with degradation, and subsequently employs this information to estimate the parameters within a given model
Type 4	Applying the filtering technique to ascertain and modify the parameters of the data-driven methodology	Filtering is a commonly employed technique in data analysis to mitigate the effects of noise and make estimations of model parameters

**Table 9 sensors-23-08124-t009:** Recent research outcomes of hybrid prognosis in aviation industry.

Authors	Domain	Methods	Findings
Singh et al. (2014) [154]	Rolling element bearing	Machine learning	The purpose of this study is to establish a framework that can be utilized as a point of reference by researchers in order to investigate potential avenues for enhancing the field of machine learning-based fault diagnosis and prognosis of renewable energy systems.
Omer et al. (2019) [191]	Clogged filter Crack propagation	Physics-based Data-driven Hybrid	The efficacy of the technique has been assessed by a comparative analysis of remaining usable life estimations derived from both hybrid and individual prognostic models. The findings indicate enhanced accuracy, robustness.
Li et al. (2022) [145]	Lug joint aluminum	Physics-based Data-driven Hybrid	This research employs an approach often utilized in sensor defect detection and proposes a novel hybrid prognostic model for a large joint aluminum. The model incorporates a bias component in both the measurement equation and the state vector. The findings demonstrate a strong and consistent pattern.
Wang et al. (2020) [172]	Wind turbine bearings	Physics-based Data-driven Hybrid	The proposed hybrid prognostics is for RUL estimation of wind turbine bearings. The exponential deterioration model is ultimately responsible for the attainment of the RUL. The findings obtained suggested methodology is both feasible and efficient in predicting the RUL of wind turbine bearings.
Zhang et al. (2023) [189]	Aircraft turbofan engine Milling	1-D convolutional neural network (1-DCNN) Bidirectional gated recurrent unit (BiGRU)	A parallel hybrid NN- 1-DCNN-BiGRU is designed for RUL. The findings provide evidence that the parallel hybrid network is capable of accurately predicting the RUL and outperform other conventional methods.
Azar et al. (2022) [190]	Aircraft engines	Reinforcement learning Machine learning Hybrid	This paper presents a unique hybrid Maintenance Decision Support System (MDSS), which has been developed by integrating ML with statistical models, semi-supervised learning, in conjunction with reinforcement learning, is utilized to optimize the cognitive behavior model.
Giannakeas et al. (2023) [192]	Composite aircraft panels	Physics-based Data-driven Hybrid	The system integrates physics-based and data-driven models in order to solve the limitations of the former and tackle concerns regarding the representativeness of the training datasets with experimental composite aircraft panel design.
Mitici et al. (2023) [193]	Turbofan engine	CNN	This paper suggests an end-to-end strategy for data-driven predictive maintenance CNN for multiple components. It also shows how data-driven predictive maintenance could help save money and improve dependability.
Cui et al. (2023) [194]	Servo actuator	Physics-based Data-driven Hybrid	Predicts the deterioration of integrated-servo-actuators (ISA) by combining a physics-based model nonlinear Wiener process (NWP) with a data-driven model Echo-State Network (ESN). The utilization of NWP is employed to characterize the physical deterioration process of the ISA. The data-driven ESN is specifically designed to optimize the description of the nonlinear degradation process of the ISA. The suggested strategy exhibits a greater level of prediction accuracy.
Faiyetole et al. (2023) [195]	Boeing 737-100	Gompertz distribution model	The study found that 737–100 had the highest survivability of all the series, while Max 8 had a high hazard ratio when interacting with the airline operator factor. It concluded that intuitive and accurate components diagnostics beyond PHM should be encouraged.

## Data Availability

Not applicable.

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
