# Peer review of "Prognostic and Health Management of Critical Aircraft Systems and Components: An Overview"

_sensors, 2023, doi:10.3390/s23198124_

Round 1

Reviewer 1 Report

This journal review paper titled "Prognostic and Health Management of Critical Aircraft Systems and Components: An Overview" provides a comprehensive overview of the field of prognostics and health management (PHM) in the context of aircraft systems. The authors effectively emphasize the importance of PHM in ensuring the safety and reliability of aircraft, which is crucial in the aviation industry. The paper offers a clear definition of PHM, its objectives, and the methods used, including predictive modeling and real-time data analysis. The inclusion of various methodologies such as physics-based modeling, data-driven techniques, and hybrid prognosis is a strong point, as it showcases the diversity of approaches in this field. Additionally, the paper highlights the need for further research in integrating hybrid PHM applications, which provides a valuable direction for future studies.

However, there are some limitations to this paper. Firstly, the absence of specific citations and references to existing literature is a drawback. It would have been beneficial to see more references to previous research in the field to support the claims made in the paper. Careful organization and integration of references within the text are crucial to maintain the paper's readability and coherence. Additionally, while the paper mentions the challenges in the implementation of PHM, it could benefit from a more detailed discussion of these challenges and potential solutions. Furthermore, the abstract lacks specific findings or conclusions, making it less clear what the reader can expect from the paper.

Reviewer 2 Report

1. What is the primary objective of Prognostic and Health Management (PHM) in aircraft systems?

2. Explain the significance of predicting the Remaining Useful Life (RUL) of subsystems in the context of PHM.

3. How does Condition-Based Maintenance (CBM) differ from traditional maintenance approaches, and how does PHM support CBM?

4. What role do predictive modeling techniques and real-time data analysis play in PHM for aircraft systems?

5. What is the key challenge identified regarding the implementation of PHM in the aviation sector?

6. Can you differentiate between data-driven techniques and physics-based modeling in the context of aviation prognostics?

7. How does PHM contribute to ensuring the safety and reliability of aircraft systems?

8. Graphical abstract can be provided to illustrate the overview of the article.

9. The quality of Figures 2 and 3 needs to be improved.

10.  Figure 9 can be enhanced by naming the inputs and output parameters.

11. Kindly change SVM to RVM in section 2.2.10.3

12. Kindly discuss the Data-driven models (DdM) and Knowledge-based models (KbM) and its challenges.

13. The authors discussed more about the theoritical part of prognostics approach. Authors are suggested to focus more on the proposed work rather than recalling the theory.

14. Give a summary and inference for each heading.

15. What are the research objectives and Hypothesis addressed in the article.

16. Give more focus on discussion part with respect to Critical Aircraft Systems and Components

17. More literatures belongs to prognostics of Aircraft Systems and Components.

18. Authors can focus on Deep learning, Reinforcement learning, Deep reinforcement laerning and Digital Twins for prognostics of Aircraft Systems

19. Suggested to give the future direction for readers to enhance the prognostics of Aircraft Systems.

20. Shorten the theory part and discuss more about the point of interest.

Moderate editing of the English language required

Round 2

Reviewer 2 Report

Congrats.

Minor editing of the English language required